# Antibiotic-Resistant *Pseudomonas aeruginosa*: Current Challenges and Emerging Alternative Therapies

**DOI:** 10.3390/microorganisms13040913

**Published:** 2025-04-16

**Authors:** Minqi Hu, Song Lin Chua

**Affiliations:** 1Department of Applied Biology and Chemical Technology, The Hong Kong Polytechnic University, Kowloon, Hong Kong SAR, China; 2State Key Laboratory of Chemical Biology and Drug Discovery, The Hong Kong Polytechnic University, Kowloon, Hong Kong SAR, China; 3Research Centre for Deep Space Explorations (RCDSE), The Hong Kong Polytechnic University, Kowloon, Hong Kong SAR, China; 4Research Institute for Future Food (RiFood), The Hong Kong Polytechnic University, Kowloon, Hong Kong SAR, China

**Keywords:** *Pseudomonas aeruginosa*, antibiotic resistance, bacteriophage, nanoparticle, antimicrobial peptides, CRISPR

## Abstract

Antibiotic-resistant *Pseudomonas aeruginosa* is a pathogen notorious for its resilience in clinical settings due to biofilm formation, efflux pumps, and the rapid acquisition of resistance genes. With traditional antibiotic therapy rendered ineffective against *Pseudomonas aeruginosa* infections, we explore alternative therapies that have shown promise, including antimicrobial peptides, nanoparticles and quorum sensing inhibitors. While these approaches offer potential, they each face challenges, such as specificity, stability, and delivery, which require careful consideration and further study. We also delve into emerging alternative strategies, such as bacteriophage therapy and CRISPR-Cas gene editing that could enhance targeted treatment for personalized medicine. As most of them are currently in experimental stages, we highlight the need for clinical trials and additional research to confirm their feasibility. Hence, we offer insights into new therapeutic avenues that could help address the pressing issue of antibiotic-resistant *Pseudomonas aeruginosa*, with an eye toward practical applications in future healthcare.

## 1. Introduction

*Pseudomonas aeruginosa* is a Gram-negative opportunistic bacterium, which was first isolated from wound pus by Gersard in 1882 [1] and which can cause infections in an immunocompromised human host [2]. *P. aeruginosa* is now one of the most widespread infections in hospitals, especially in intensive care units (ICUs), and is increasingly difficult to treat due to its high resistance to common antibiotics. It is a leading cause of nosocomial infections, accounting for approximately 7.1–7.3% of all HAIs [3], with a higher prevalence in intensive care units (ICUs), where it is responsible for up to 16.2% of infections. *P. aeruginosa* is associated with a number of adverse outcomes, including ventilator-associated pneumonia (VAP), surgical site infections, urinary tract infections (UTIs), and bloodstream infections (BSIs) [3]. In severe cases, such as VAP and BSIs, the mortality rate can range from 32% to 58.8% [3]. In cystic fibrosis patients, chronic *P. aeruginosa* infections are a major contributor to morbidity and mortality, with prevalence rates as high as 49.6% in some populations [3].

Moreover, urinary tract infections (UTIs) caused by *P. aeruginosa* pose a significant clinical challenge, particularly within healthcare settings, due to their association with catheter use, urinary tract abnormalities, and immunocompromised states [4]. The intrinsic and acquired resistance mechanisms of pathogens, including biofilm formation and the capacity to invade bladder epithelial cells, contribute to the complexity of treatment regimens for these infections. *P. aeruginosa* is a major cause of catheter-associated urinary tract infections (CAUTI), and biofilms on catheter surfaces complicate eradication [4].

Antimicrobial resistance (AMR) has been identified as one of the top ten public health threats worldwide. In 2019, it was estimated that 1.27 million people died directly from AMR, and a further 4.95 million people died from drug-resistant infections [5]. The phenomenon of antimicrobial resistance is undermining the effectiveness of antibiotics, which are the cornerstone of modern medicine. As antimicrobial resistance spreads, the risks associated with common infections, surgical procedures and treatments such as chemotherapy and organ transplants increase dramatically [6]. The impact of *P. aeruginosa* is further exacerbated by its intrinsic and acquired AMR, making it a critical public health concern [7,8]. The prevalence of multidrug-resistant (MDR) and extensively drug-resistant (XDR) strains has escalated, with resistance rates to carbapenems exceeding 30% in certain regions [9]. MDR refers to the resistance to multiple antimicrobial agents within three or more classes of antibiotics, while XDR refers to resistance to almost all antimicrobial agents across multiple antibiotic classes, with susceptibility to only one or two categories of antibiotics [10]. Typically, XDR pathogens are of greater concern than MDR pathogens because they leave clinicians with very few, if any, effective treatment options [11]. This resistance complicates treatment, resulting in prolonged hospital stays, escalated healthcare costs, and increased mortality.

Carbapenem-resistant *P. aeruginosa* (CRPA) is of particular concern because carbapenems are often considered the antibiotic of last resort for treating serious infections [12]. In some regions, *P. aeruginosa* has been reported to exhibit resistance to carbapenem antibiotics to exceed 30%, a development which severely limits treatment options [3]. Within the United States, *P. aeruginosa* is responsible for approximately 51,000 hospital acquired infections per year, with infections due to CRPA leading to increased morbidity, mortality and healthcare costs. The emergence of MDR and XDR *P. aeruginosa* strains would lead to worse clinical outcomes. For instance, bloodstream infections caused by MDR *P. aeruginosa* have been shown to have a high mortality rate of 58.8%, compared to 43.2% for non-MDR strains [13,14]. Measures to combat *P. aeruginosa* AMR include antimicrobial stewardship programs, improved infection control measures and the development of new antibiotics.

Hence, the battle against antibiotic-resistant *P. aeruginosa* highlights the urgent need for innovative antimicrobial solutions. Firstly, the limited efficacy of conventional antibiotics against resistant strains pushes the need to find alternative treatments [15]. Furthermore, the inadequacy of the drug development pipeline is also contributing to the failure to generate effective treatments, compared with the traditional antibiotic treatment [16]. Hence, researching novel alternative therapeutic approaches is essential to overcoming this resistance and enhancing the efficacy of treatments [17].

In our review, we will explore various anti-*Pseudomonads* alternative therapies, including antimicrobial peptides, quorum sensing inhibitors (QSIs) and nanoparticles. Novel alternative therapies including bacteriophage therapy and CRISPR-Cas system will also be discussed. Moreover, these new approaches can also help maintain the potency of currently available antimicrobials and prevent the development of new antibiotic resistance [18]. These efforts will promote innovation in antimicrobial therapy and improved our management of antibiotic resistance in clinical practice.

Our review will discuss recent articles, including peer reviewed research articles and other reviews from the last 10 years (2014–2024). Clinical trials and case–control articles are included, whereas website articles and other media resources are excluded. All the studies are found in PubMed and Google Scholar, using specific keywords related to Pseudomonas aeruginosa, biofilm formation, antibiotic resistance, and alternative therapies including bacteriophage therapy, nanoparticles therapy and CRISPR-Cas system.

## 2. Mechanisms of Antibiotic Resistance in *Pseudomonas aeruginosa*

Several mechanisms contribute to the development of antibiotic resistance of *P. aeruginosa*, including intrinsic resistance, acquired resistance and adaptive resistance.

### 2.1. Intrinsic Resistance

An intrinsic resistance to *P. aeruginosa* can reduce or inactivate the effectiveness of antibiotics through their own products or functional properties, via [1] the possession of efflux pumps that exclude antibiotics from the bacterial cells, and [2] low outer membrane permeability.

#### 2.1.1. Active Transport Mode: Efflux Pumps

Efflux pumps play crucial roles in developing the intrinsic antibiotic resistance in *P. aeruginosa*. Their ability to export the antibiotics provide *P. aeruginosa* with extra survival chances under antibiotic choice stress. Efflux pumps can pump specific substrates, including various classes of antibiotics (e.g., fluoroquinolones, beta-lactams, tetracyclines) and other toxic molecules across the cell membrane and out of the bacterial cells [19], with the use of ATP or proton motive force [20]. The expulsion of these substrates from the cell results in a reduction in their intracellular concentration and prevents them from reaching their target sites, including ribosomes, DNA gyrase, and cell wall synthesis machinery [19]. Efflux pumps can also confer low-level resistance, which can facilitate the survival of bacteria in sub-lethal concentrations of antibiotics, thus providing an opportunity for the acquisition of additional resistance mechanisms [21]. Contributing to biofilm formation, efflux pumps also enhance resistance by creating a physical barrier that limits antibiotic penetration and protects the bacterial community [22].

*P. aeruginosa* possesses a number of efflux pumps that actively expel antibiotics out of the cell, which includes the ATP-binding cassette (ABC) superfamily, the major facilitator superfamily (MFS), the resistance-nodule-dissection (RND) family, the small-molecule multidrug-resistant (SMR) family, the multidrug- and toxic-compound-excreting (MATE) family, and the recently described Aspergillus antimicrobial compound efflux (PACE) family [23]. The most important and well-studied of these proteins is the RND family, including the core drug-resistant efflux pump genes of P. aeruginosa, such as bla, ampC, and the genes encoding multidrug efflux pumps (Mex), such as MexXY, MexAB-OprM, MexCD-OprJ, and MexEF-OprN [24,25,26]. In clinical isolates of P. aeruginosa, MexXY-OprM can excrete aminoglycosides, whereas the highly conserved MexAB-OprM is mainly responsible for the exocytosis of β-lactams and quinolones [27]. Next, MexCD-OprJ is responsible for the exocytosis of β-lactams, while MexEF-OprN is capable of exocytosis of quinolones [26].

On the other hand, the MFS pumps are characterized by the transportation of a more limited range of substrates, including tetracyclines and chloramphenicol [28]. The SMR family pumps, which are smaller in size, facilitate the expulsion of smaller cationic compounds, such as quaternary ammonium compounds and dyes [29]. The ABC family, though primarily involved in nutrient transport, can also participate in antibiotic resistance [30]. Finally, the MATE family pumps, such as PmpM, use proton or sodium ion gradients to extrude substrates like fluoroquinolones and other cationic compounds [31]. The synergistic activity of these efflux pump families enables *P. aeruginosa* to achieve a high level of MDR, making it a challenging pathogen to treat in clinical settings. These efflux pumps greatly increase the antibiotic resistance of *P. aeruginosa*, where the number and type of efflux pumps have a direct positive correlation with the drug resistance profile of *P. aeruginosa* [32]. Hence, efflux pumps are potential targets of inhibitors to reduce antibiotic resistance [33].

#### 2.1.2. Low Membrane Permeability to Antibiotics

Since antibiotics need to pass through the outer membrane of the bacteria to act on the internal target [17], Gram-negative bacteria are generally more antibiotic resistant than Gram-positive bacteria. Notably, the outer membrane of *P. aeruginosa* is 12–100 times less permeable to antibiotics compared to other Gram-negative bacteria such as *E. coli* [34]. This is attributed to an OprF channel protein that opens less than 5% of the surface of the outer membrane, with a very low permeability to antibiotics [35]. *P. aeruginosa* has several specific pore proteins, and the other channel proteins include OprB, OprD, OprP, and OprO [34], which are responsible for the entry and exit of different antibiotics through the outer membrane [34].

### 2.2. Acquired Resistance

Acquired resistance is another approach by which *P. aeruginosa* acquires antibiotic resistance, including horizontal gene transfer and accumulation of genetic mutations.

#### 2.2.1. Horizontal Gene Transfer

Bacteria can acquire antibiotic resistance genes from other bacteria via horizontal gene transfer, including transformation [36], transduction [37], and conjugation [38]. Antibiotic resistance genes can be found on plasmids, phages, transposons, and integrons [39].

Plasmids are key vectors for the spread of β-lactams, aminoglycosides, and fluoroquinolones resistance via conjugation [18]. Next, transposons, such as Tn21, facilitate the mobilization of resistance genes by integrating into plasmids or the chromosome, while integrons capture and express resistance gene cassettes such as *aadA* and *dfrA* [40]. Lastly, genomic islands, such as PAGI-1 and PAPI-1, can also harbor antibiotic resistance and virulence genes, where they integrate into the chromosome and spread via conjugation or transduction [40]. For example, the exogenous acquisition by wild-type bacteria of the β-lactamase gene confers bacterial resistance to β-lactams [41]. The common genes and plasmids associated with AMR in *P. aeruginosa* are listed in Table 1.

#### 2.2.2. Mutation-Accumulated Antibiotic Resistance

Mutation-accumulated antibiotic resistance is caused by mutations in the genetic material [52], as the result of the adaptation of bacteria to antimicrobial molecules. Firstly, the emergence of *P. aeruginosa* resistance to colistin is a major concern [53]. Heterogeneous resistance to colistin in *P. aeruginosa* involves the up-regulation of the LPS modification system and mutations in specific gene loci for lipid A synthesis which would modify lipid A by 4-amino-4-deoxy-L-arabinose (L-Ara4N) [54]. Other two-component regulatory systems (TCS) such as *PhoPQ, ParRS* and *CprRS* are also involved in mediating colistin heterogeneous resistance [55].

Next, carbapenem resistance is an emerging concern for *P. aeruginosa* infections [56]. Out of 382 carbapenem-resistant *P. aeruginosa* strains and 148 genes sequences isolated in the clinic, 87.1% of carbapenem-resistant *P. aeruginosa* had inactivation of the *oprD* gene, and 28.8% showed mutations in the genes regulating the MexAB-OprM efflux pumps (*mexR*, *nalC* and *nalD*) [57], suggesting that *P. aeruginosa* exhibits significant genetic diversity in the resistance against carbapenems.

### 2.3. Adaptive Resistance

Adaptive resistance is a reversible resistance that involves transient changes in the regulatory expression of genes and proteins in bacteria exposed to a hostile environment for a long period of time, thereby enhancing their ability to resist in antibiotic-containing environments [58]. Two adaptive resistance mechanisms, including formation of biofilm and emergence of persister cells are now studied.

#### 2.3.1. Formation of Biofilm

Biofilm is a heterogeneous structure consisting of a microbial community and the extracellular polymeric substance (EPS) that they secrete [59]. The role of biofilms in antibiotic resistance is attributable to their physicochemical properties. Polysaccharides, including alginate, Psl, and Pel, provide structural integrity and adhesion; extracellular DNA (eDNA) stabilizes the matrix and neutralizes antibiotics; proteins promote cohesion; and lipids increase hydrophobicity [60,61]. The viscous nature of the EPS reduces the penetration and utilization efficiency of antibiotics [62], so biofilms are more resistant to antibiotics than planktonic cells [63]. Moreover, the EPS acts as a physical barrier which harbors metabolically inactive persister cells, promotes horizontal gene transfer of antibiotic resistance genes, and protects bacterial cells from predation and detection of bacterial-secreted metabolites by predators [64,65,66].

Of all the EPS components, eDNA plays an important role in the development of bacterial resistance. The eDNA is released by cell lysis and reactive secretion [67]. In addition to suppressing the innate immune response [68], eDNA played a role in the EPS in support of biofilm stabilization [69], genome integration [70], chelation with metal cations and induction of antibiotic resistance formation [71].

By binding to divalent metal cations in the environment, *P. aeruginosa* exhibits resistance to cationic antimicrobial peptides (CAPs) and aminoglycosides [72]. The former resistance was due to activation of the *PhoPQ* and *PmrAB* systems [73,74], while the latter was due to the binding of positive charged aminoglycosides to negatively charge eDNA [72]. Interestingly, eDNA can bind to specific exopolysaccharides such as Psl and Pel [75] to form DNA-polysaccharide fibrillated structures [76], enabling biofilm formation and enhancing biofilm protection against antimicrobials [23].

*P. aeruginosa* biofilm formation is regulated by several interconnected signaling systems, including the quorum sensing (QS) system, the two-component regulatory systems (TCS) GacS/GacA and RetS/LadS, and the c-di-GMP signaling system [77].

Firstly, four QS systems have been identified: *las*, *rhl*, *pqs* and *iqs* [78].The function of QS is to regulate population dynamics through the secretion of autoinducers (AIs). In the *las* system, LasI and LasR are mainly responsible for the synthesis and detection of N-(3-oxododecanoyl)-L-homoserine lactone (3-oxo-C12-HSL), respectively. This autoinducer activates genes involved in biofilm formation, including those encoding EPS components [79]. The *rhl* system involves the transcription factor RhlR, which, upon reaching a threshold concentration, facilitates the expression of the lasI gene, in turn promoting the production of virulence factors, including LasA protease, LasB elastase, Apr alkaline protease, and exotoxin A. RhlI and RhlR are primarily responsible for the synthesis and detection of N-butanoyl-L-homoserine lactone (C4-HSL), which also promotes the transcription of RhlI, which in turn promotes the expression of virulence factors [79]. Meanwhile, Rhl system regulates the production of rhamnolipids, which are essential for biofilm structuring and dispersal [80]. The *pqs* system generates 2-heptyl-hydroxy-4-quinolone, which has the capacity to interact with the acylhomoserine lactone (AHL) system in a complex manner, thereby functioning as an intermediate hub between the *las* and *rhl* systems [81].

C-di-GMP, a small intracellular molecule that is the second messenger of the bacterial metabolic pathway, plays a pivotal role in regulating the growth of biofilm [82]. The high levels of c-di-GMP contribute to the formation of biofilms, whereas low levels of c-di-GMP lead to biofilm dispersal and the formation of biofilm-dispersed cells [83,84,85,86,87].

#### 2.3.2. Formation of Persister Cells

Persister cells located inside the biofilm develop adaptive resistance to the effects of long-term antibiotics (Figure 1). These cells are not genetically antibiotic resistant but can tolerate high concentrations of antibiotics [88], because they are in a near-dormant state [89]. Their metabolic inactivity and the shutdown of the synthesis of antibiotic-acting molecules are properties that render these cells hyper-resistant to high concentrations of antibiotics [90]. They will resume growth when external conditions in the absence of antibiotics are favorable.

Moreover, the toxin–antitoxin system is involved in the formation of holding cells and induces dormancy [91]. The toxin–antitoxin system usually consists of two gene parts in a manipulator, one responsible for encoding a stable toxin that disrupts life processes, such as mRNA translation, and the other responsible for encoding an unstable toxin such as RNA or protein that prevents toxicity [92]. It is also related to the bacterial survival in adversity, including the activation of persistent cell formation, and control of programmed cell arrest or death [93].

With the above-mentioned mechanisms, MDR and XDR *P. aeruginosa* strains renders limited choice of antibiotics in their treatment. Clinically, MDR and XDR *P. aeruginosa* can lead to longer and more complicated infections [94], even playing a role in tumor biology as intra-tumor biofilms [95,96,97,98], so it is necessary for both patients and society to invest more time money and effort to achieve better treatment outcomes. The development of new antibiotics is time-consuming and tedious [99], which warrants the development for novel alternative antimicrobial therapies which can eliminate antibiotic-resistant *P. aeruginosa* and disperse its associated biofilms [100,101].

### 2.4. Current First-Line Antibiotics Used to Treat MDR Pseudomonas Infections

The treatment of MDR *P. aeruginosa* infections poses significant challenges due to its intrinsic and acquired resistance mechanisms, so several new antibiotic combinatorial treatments have emerged as first-line options for treating such infections. Cefazolin–tazobactam, a novel combination of cephalosporins and β-lactamase inhibitors, has been shown to be highly effective against MDR and carbapenem-resistant *P. aeruginosa* [102], particularly in complicated intra-abdominal infections [102], urinary tract infections [103] and hospital-acquired pneumonia [7]. Cefozoxime-avibactam, another beta-lactam–beta-lactamase inhibitor combination, was effective against broad-spectrum beta-lactamase-producing and carbapenem-resistant strains, with a susceptibility rate of over 95% against *P. aeruginosa* [104]. A glycosidophilic cephalosporin, cefiderocol, exploits the bacterial iron transport system to overcome resistance mechanisms such as exocytosis pumps and pore-protein mutations, maintaining its activity against 88.7% of *P. aeruginosa* isolates [105]. Finally, polymyxin antibiotics, such as colistin, remain the last option for the treatment of severely and XDR infections [106], but their use is limited by significant toxicity and emerging colistin resistance [107]. Consequently, there is an urgent need for effective therapies against drug-resistant *P. aeruginosa* strains that are less toxic and have a broader range of action than conventional antibiotics.

Prevention strategies also shed light as a potential treatment against *P. aeruginosa* infections. In the treatment of uncomplicated UTIs in children, short courses of antibiotic therapy (5–7 days) have been shown to be comparable in efficacy to the standard 10-day to 14-day course of therapy [108]. These findings are particularly salient in light of the concomitant reduction in antibiotic exposure, development of resistance, and adverse effects [108]. A meticulous examination of randomized controlled trials reveals no statistically significant disparity in recurrence rates between short courses and standard therapy, thereby substantiating the efficacy of this approach in uncomplicated UTIs caused by *P. aeruginosa* [109]. However, in chronic diseases such as cystic fibrosis, where infections are often recurrent and associated with biofilms [110], short courses of therapy are less effective and require extended or suppressive antibiotic regimens [111].

## 3. Current Alternative Therapeutic Approaches

### 3.1. Phage Therapy

#### 3.1.1. Overview of Bacteriophages Targeting *P. aeruginosa*

Phage therapy is gaining increasing attention as a potential option for the treatment after traditional antibiotic treatments failed to eliminate bacteria. Numerous studies have demonstrated the efficacy and eukaryotic tolerance of phage therapy against bacteria in in vitro and in vivo conditions [112,113]. Specific treatments are targeted against urinary tract infection [114], open wounds [115] or implants [116], and pulmonary inflammation in clinical trials [116].

*P. aeruginosa* infection has attracted strong attention as a target for phage therapy. The first phage strain targeting *P. aeruginosa* was isolated from sewage around the 1950s [117], which brought new hope for the treatment of this highly antibiotic-resistant bacteria [118]. Other common *P. aeruginosa* phages include PB-1, M-1, phiKZ, and LUZ24 [119]. The current attempts of phage therapy have yielded encouraging results, where some phage mixtures can even penetrate the *P. aeruginosa* biofilm and disrupt the biofilm structure by inducing the synthesis of enzymes such as polysaccharide depolymerase [120] (Figure 2).

Moreover, phage cocktails can better disrupt *P. aeruginosa* cells due to their ability to produce peptidoglycan hydrolases, or endolysins to lyse the pathogen [121]. Alternatively, phages can eradicate *P. aeruginosa* biofilms by directly inhibiting biofilm synthesis [121]. Furthermore, by inducing the synthesis of quorum-quenching (QQ) lactonase through genetic modification, phages can even hydrolyze acyl homoserine lactones (AHL) and inhibit QS activity to inhibit *P. aeruginosa* biofilm formation [122].

#### 3.1.2. Advantages and Limitations of Phage Therapy

The utilization of phages in the antimicrobial therapy of *P. aeruginosa* has several advantages.

The rationale behind utilizing phage therapy is multi-faceted. Phages exhibit a high degree of penetration and specificity in comparison with conventional antibiotic therapies, which is useful in personalized treatment [123]. Furthermore, phages do not produce harm to human cells and resident microbes, thus achieving the effect of killing the target bacteria without interfering with the normal microbiota in the human body [124]. Next, phages are effective against MDR and XDR strains, providing a viable alternative where antibiotics fail. A further advantage of phage therapy is that some phages can penetrate and disrupt biofilms, via the production of EPS-degrading depolymerases [125]. Additionally, phages are able to self-replicate at the site of infection, ensuring elevated local concentrations, and are self-limiting when they are cleared once the bacterial host is eliminated [124]. Once the infection site is reached, the number of phage particles increases exponentially, thereby achieving the desired therapeutic effect without the need for repeated applications of phage. As the bacteria are killed, the titer of the phage decreases until the phage is completely eliminated from the patient’s body [126].

For instance, the phage PELP20 is highly effective in eradicating *P. aeruginosa* strains isolated from cystic fibrosis (CF) patients in an in vitro artificial sputum medium biofilm model (LESB65), and significantly enhances bacterial clearance in a mouse model of chronic lung infection [127]. Even the pretreatment of hydrogel-coated catheters with phage M4 can significantly reduce *P. aeruginosa* biofilm formation [128].

Other than their inherent nature to prey on bacteria, phages can be genetically engineered to possess additional functions, thus improving therapeutic efficacy [128]. In a lysogenic phage-based genetic modification, the *P. aeruginosa* lysogenic phage Pf3 gene was modified into a non-lysogenic, non-replicative phage, Pf3R, by replacing the output protein gene with the BglII restriction endonuclease gene [129]. This modified phage has been shown to kill *P. aeruginosa* PAO1 in vitro with the same efficiency as normal Pf3, but with a significantly reduced LPS release, thereby reducing the activation of LPS-induced host immune response. Furthermore, phage therapy can be combined with antibiotics to enhance antimicrobial treatment against pathogens and their biofilms. For instance, the combination of phage (ATCC 12175-B1 (Pa1), ATCC 14203-B1 (Pa2), and ATCC 14205-B1 (Pa11)) and ciprofloxacin on *P. aeruginosa* biofilms has been shown to have a superior removal effect compared to the individual treatment [130].

However, phage therapy has not become a mainstream therapeutic modality yet, because of several primary clinical challenges. As a new therapy under development, the primary limitations identified for this system include a restricted host range, necessitating precise strain matching; inadequate standardized protocols in managing pharmacodynamics (PD) and pharmacokinetics (PK) [131], and emergence of phage resistance. For PD, the metabolism and elimination of phages are influenced by various factors, including the immune system and size of the host, and the host’s own microbiota [132]. This poses a significant challenge in terms of maintaining effective phage concentrations and preventing rapid elimination by the host [133]. It is crucial to evaluate the effective infection concentration, the active form of functioning phages, and the reaction by the immune system [134].

For PK, the clinical challenge is how to effectively deliver the phage to its target location while preserving a high degree of activity [135]. The existing routes of phage administration are highly varied, including oral, topical, intravenous, intra-operative, intra-rectal and nebulization, with different states of formulation (gas, liquid and solid) [136,137]. Moreover, the phage metabolism and its inactivation are influenced by the environmental pH and immune responses, such as phagocytosis of phages by liver-based Kupffer cells and production of phage-neutralizing antibodies in the spleen [138]. Meanwhile, host organs, especially the human kidneys, also contribute to phage clearance by regulating phage load in vivo [131]. Furthermore, discrepancies in phage excretion via urine have been observed to be contingent on the age of the patient [131,139]. Consequently, the selection and commercialization of suitable phages represents a significant challenge in the widespread adoption of phage therapy. Phage therapy may also cause potential side effects, where bacterial lysis by phages can release bacterial fragments that can cause inflammation, activate immune cells, and ultimately lead to systemic organ failure (known as the Jarisch–Herxheimer effect) [135].

Lastly, *P. aeruginosa* can rapidly develop phage resistance, while phages can also evolve in response, thus introducing a degree of uncertainty and instability regarding the phage’s effectiveness as a pharmaceutical agent. Experimental co-evolution of *Pseudomonas fluorescens* with its phage resulted in an acceleration of the bacterium’s mutation rate [140]. The predominant mutations observed are deletions of genes, including those encoding LPS, biofilm-associated EPS, and T4P receptors, leading to the modification of phage attachment or adsorption sites [141]. Moreover, clinical infections are often multispecies, so a phage cocktail therapy combining different phage species emerges as a more efficient option [142]. However, the process of designing phage mixtures is considerably more intricate than that of creating combination antibiotic regimens, necessitating substantial manpower and resources for future applications in personalized therapy.

#### 3.1.3. Clinical Trial Status

As early as 2009, phage cocktail therapy was employed clinically for the purpose of postoperative infection clearance in patients suffering from burns or open wounds. A mixture comprising approximately 82 phage (Myoviridae), 10 phage ISP (Myoviridae) against *Staphylococcus aureus* and PNM (Podoviridae) against *P. aeruginosa* was utilized, achieving a successful reduction in the infection condition of eight burn patients. The potential of phages (pp1131) in the treatment of *P. aeruginosa* endocarditis has also been demonstrated in a mouse model [143].

Next, a male patient with a chronic MDR *P. aeruginosa* infection in a Dacron aortic arch prosthesis, unresponsive to three years of antibiotics, successfully managed the infection with OMKO1 phage therapy, which disrupted biofilms and restored bacterial susceptibility to antibiotics, significantly enhancing treatment efficacy [144]. In 2019, a 46-year-old male with a chronic antibiotic-resistant *P. aeruginosa* bone infection achieved long-term remission with two weeks of PASA16 phage therapy combined with ceftazidime, experiencing only minor side effects over four years [145]. Moreover, an elderly woman with a prosthetic joint infection caused by *P. aeruginosa* was successfully treated with Pa53 phage and meropenem, with no recurrence or serious adverse effects observed over two years, highlighting the therapy’s safety and efficacy [146]. A 57-year-old woman with non-CF bronchiectasis and MDR *P. aeruginosa* showed clinical improvement without adverse events after four weeks of intravenous AB-PA01 phage therapy combined with inhaled colistin, highlighting its potential for treating complex resistant infections [116]. These cases collectively emphasize the promise of phage therapy in treating chronic, antibiotic-resistant infections, particularly when combined with antibiotics.

### 3.2. Antimicrobial Peptides (AMPs)

AMPs are a class of small-molecule proteins, typically comprising 15–20 amino acid residues. To date, 3100 antimicrobial peptides has been identified from diverse sources, including bacteria, soil, and animals [147,148]. AMPs possess potent antibacterial activity against bacteria, fungi and viruses, with low toxicity to eukaryotic cells, good thermal stability, solubility and low tendency for bacteria to develop resistance [149].

#### 3.2.1. Mechanisms of Action Against *P. aeruginosa*

Each AMP has a distinct mechanism of action depending on the target, and the most common types can be classified as acting on the cell wall, cell membrane, intracellular target receptors, and on biofilms [150]. For *P. aeruginosa*, most AMPs act on cell membranes and biofilms. Due to the negative charge of these membrane proteins, they have a natural tendency to bind to AMPs, which carry a positive charge [60]. Examples of AMPs in this category include alamethicin [151,152] and pardaxin [153,154].

In the toroidal pore model, AMPs interact with phospholipid heads to bend the membrane, forming annular pores that release bacterial contents and lead to cell death [155]. Representative AMPs in this category include Lacticin Q and bee toxin [156,157]. In addition to the others AMPs, there are also non-cavity models, such as the carpet model [158] and the polymeric model [159], which do not kill bacteria by punching holes in the cell membrane. Other than targeting the cell membrane, AMPs can inhibit QS [160,161], bacterial adhesion [160,162], biofilm formation [163], and disrupt the biofilm matrix [164]. For example, LL-37 can inhibit rhamnosyltransferase expression and reduces biofilm production by acting on the *rhl* QS system [160,165], whereas Esculentin-1a disrupts *P. aeruginosa* biofilms by disrupting the cell membrane [166]. Hence, the mechanisms of action of AMPs are very diverse (Table 2), and their understanding is still a work in progress.

#### 3.2.2. Challenges in Stability, Delivery, and Toxicity

Although AMPs hold great promise for the treatment of bacterial infections, the clinical use is encumbered by several major challenges, the most significant of which pertain to their susceptibility to host and bacterial proteolytic enzymes, thereby affecting their stability and diminishing their therapeutic half-life [177]. In addition, despite their selectivity for bacterial membranes, certain AMPs exhibit haemolytic activity, which raises safety concerns. Thirdly, while the probability of bacterial resistance to antimicrobial peptides is comparatively lower than that of conventional antibiotics, *P. aeruginosa* may elicit resistance through mechanisms including membrane modification, efflux pumps and protease production [178,179].

A further challenge is posed by the high production cost of synthetic AMPs, especially those with complex structures or those which contain unnatural amino acids [180]. Expression of AMP genes in microbes by genetic engineering may lead to self-killing of the host microbe [165]. Furthermore, AMPs frequently demonstrate suboptimal pharmacokinetic properties, such as rapid systemic clearance and limited bioavailability, necessitating the development of advanced delivery systems to enhance their efficacy [181]. Additionally, the host immune system may recognize AMPs as foreign substances, which could result in immune clearance or inflammatory responses [182].

### 3.3. Quorum Sensing Inhibitors (QSIs)

#### 3.3.1. Types of QSIs

QSIs disrupt bacterial communication by targeting key components of the quorum sensing (QS) system, which the bacteria use to coordinate group behaviors [183]. The inhibition of QS can be achieved through various approaches: (i) inhibition of the synthesis of QS signaling molecule, (ii) degradation of the QS signaling molecule, (iii) competition of QSI with the signaling molecule over the receptor binding site, (iv) inhibition of QS genes by preventing the signaling molecule from binding to their promoters, and (v) scavenging of a QS-signaling molecule via macromolecules such as antibodies and cyclodextrins [184,185]. In terms of their chemical nature, QSIs can be classified as natural products, synthetic molecules and antibodies (Table 3). Since QSIs do not kill the target bacteria, they are typically used in combination with antibiotics to improve antibiotic efficacy [186].

#### 3.3.2. Challenges in Developing QSIs, Including Specificity and Potential Resistance

A significant challenge in the development of QSI arises from the gradual evolution of resistance to QSI in bacteria [208]. For example, the bromofuranone QSI is actively excreted from *P. aeruginosa* [209], demonstrating that active exocytosis of QSI is one of the potential mechanisms of resisting a QS disruption. Moreover, bacterial elimination cannot be fully achieved without the existence of bacteria-killing antibiotics [210]. Lastly, the selectivity of QSI against specific QS systems in each bacterial species renders the development of broad-spectrum QSIs difficult.

Although the concept of using QSI to eliminate pathogenic bacteria has been proved to be feasible in laboratory level, translating the therapy into clinical trials met several problems. In clinical trials, not all isolated bacteria strains are sensitive to QSI, especially those with MexAB-OprM efflux pump mutation [211]. Such mutations are frequently observed in clinical isolates as a consequence of the intensive antibiotic therapy to which these isolates are subjected [212]. Moreover, bacteria with QS mutations may express higher pathogenic traits [209,213,214], such as more biofilm formation [215] and β-lactamase activity [216]. Even in *P. aeruginosa* intubation infection, the use of azithromycin as a QSI may contribute to selecting more virulent wild type instead of less virulent *lasR* mutants [213].

In chronic infections such as cystic fibrosis (CF) and COPD, *lasR* mutants of *P. aeruginosa* frequently emerge. These mutants can evade the host immune response and reduce energetically costly virulence factors such as proteases and siderophores, thereby gaining a competitive advantage [217]. These mutants exhibit metabolic flexibility in carbon catabolite repression, cross-feeding and constitutively express biofilm-adapted genes, enhancing survival in host environments [218]. The ability of lasR mutants to modulate immune responses and adapt to biofilms underscores their role in persistent infections and negates the effectiveness of *lasR*-based QSIs [219].

The pathogen has the capacity to activate alternative signaling systems, such as the *pqs* QS, in order to compensate for inhibited QS circuits. Furthermore, *P. aeruginosa* may produce enzymes (e.g., oxidoreductases) that degrade or modify QSIs, rendering them ineffective [220]. Phenotypic heterogeneity within bacterial populations has also been demonstrated as a factor that can contribute to resistance, where QS-deficient “cheater” cells exploit QS-dependent public goods, such as virulence factors, without contributing to signal production, thereby reducing the selective pressure imposed by QSIs [220].

### 3.4. CRISPR-Cas Systems for Targeted Gene Editing

#### 3.4.1. Concept of Using CRISPR to Disrupt Resistance Genes

There are two distinct CRISPR-Cas system-based therapeutic interventions against drug-resistant pathogenic bacteria, namely the pathogen-centric therapy and the gene-centric approach. The pathogen-centered therapeutic strategy involves the direct targeting of the bacterial chromosome, leading to the destruction of the bacteria by cutting off their genetic material [221]. The ability of CRISPR-Cas9 to specifically inactivate target genes on the bacterial chromosome has been demonstrated in the cases of *Staphylococcus aureus* [222,223], *Clostridioides difficile* [224,225], and *Klebsiella pneumoniae* [226] (Table 4 and Table 5).

For gene-centric approach, this is achieved by either the CRISPR-Cas9 system or CRISPR-Cas13a system. The former can recognize and excise specific nucleic acid sequences on chromosomes [227], while the latter has the capacity to eliminate bacteria containing specific sequences on both chromosomes and plasmids [228]. Furthermore, the CRISPR-Cas13a system does not directly cut bacterial DNA, as it targets bacterial mRNA with low mutation activity [228].

**Table 5 microorganisms-13-00913-t005:** Mechanism of pathogen-centered CRISPR therapy for *E. coli, E. faecalis* and *K. pneumoniae*. Only the most common bacterial species are selected and the order of the bacteria was listed in alphabetical order.

Bacteria	Gene	Mechanism	Reference
*E. coli*	*Mcr-1*	Insert CRISPR Cas9 to pCas9 plasmid to resensitize bacteria to Colistin	[229]
*E. faecalis*	*ermB*, *tetM*	A constitutively expressed CRISPR/Cas9 was designed using pD1 to reduce antibiotic resistance	[230]
*K. pneumonia*	*ramR*, *tetA*, *mgrB*	Use CRISPR Cas9 to modify pSGKP-spe and pBECKP-spe plasmids to make bacteria sensitive to antibiotics (e.g., Colistin)	[226]

The CRISPR-Cas system has been used to excise drug-resistant genes from drug-resistant bacteria and restoring their susceptibility to antibiotics [231]. For example, the modified CRISPR-Cas9 system can target and eliminate colistin resistance genes, such as mcr-1, in *E. coli* and *K. pneumoniae*, thereby restoring bacterial susceptibility to colistin [232]. Next, CRISPR-Cas systems can restrict the transfer and maintenance of blaKPC-harboring plasmids in *K. pneumoniae*, thus reducing the spread of carbapenem resistance in clinical settings [233]. Even in Gram-positive bacteria, CRISPR-Cas system can eliminate the resistance of *S. aureus* to nacarbamycin and methicillin, achieving a removal rate of 99.99% for high-copy plasmids in bacteria [234].

Next, Rodrigues and colleagues [230] successfully modified the splice plasmid pPD1 using an intact, constitutively expressed CRISPR-Cas9 system, thus enabling the excision of erythromycin (ermB) and tetracycline (tetM) resistance genes present within the plasmid, resulting in a notable reduction in antibiotic resistance both in vivo and in vitro. Interestingly, CRISPR-Cas system can be used in combinatorial therapy with nanoparticles, where nanoparticles function as carriers to transport the CRISPR-Cas protein-nucleic acid complex into *P. aeruginosa* [235].

#### 3.4.2. Progress and Challenges in Clinical Application

The CRISPR-Cas system has demonstrated considerable potential in addressing drug-resistant bacteria. However, the use of a solitary, non-essential target for remediation proves inadequate when confronted with the complexity of MDR plasmids. The employment of CRISPR arrays or the utilization of multiple sgRNAs, in conjunction with the design of CRISPR-Cas systems targeting essential genes on drug-resistant plasmids, can enhance the efficiency of CRISPR-Cas-based cleanup processes [236].

Next, the efficiency of delivery of the large 160 kDa protein-RNA complex to the site of the pathogen and the subsequent exertion of antimicrobial effects remain a significant challenge. Existing literature proposes a range of methods for delivery, including the use of plasmid vectors [237], mild phages [234], nanoparticles [238], and electroporation [239]. However, these methods face challenges such as low stability, delivery efficiency, and potential immunological risks [238]. Moreover, CRISPR-Cas therapies are confronted with other technical challenges, including off-target effects, PAM sequence requirements [240], complex diversity of microbial communities; and even evolved resistance to the CRISPR Cas system [236]. Addressing these challenges is imperative for the widespread adoption and successful implementation of this technology.

### 3.5. Nanoparticles and Drug Delivery Systems

Nanoparticles with a diameter smaller than 100 nm have the capacity to function as carriers for antimicrobial agents [241]. The bactericidal effect of nanoparticles can be categorized into two distinct classifications. Firstly, there are nanoparticles that possess intrinsic bactericidal properties, including nitric oxide releasing nanoparticles, chitosan-containing nanoparticles, and metal-containing nanoparticles [242,243,244,245]. Secondly, nanoparticles can augment the bacteriostatic effect of existing bacteriostatic drugs through combination therapy. In comparison with traditional antibiotic therapy, which requires higher effective concentrations, nanoparticles can carry drugs with lower overall concentrations but higher localized concentrations [246]. Moreover, the ability of nanoparticles to target antimicrobial drugs to the site of infection enables a higher dose of the drug to be administered directly at the site of infection [247], thus overcoming existing resistance mechanisms while reducing adverse effects on the patient [248]. The utilization of nanoparticles in this manner allows them to traverse through biofilms which are typically resistant to traditional antibiotics and directly reach the bacteria, thereby exerting a more potent bactericidal effect [249].

The encapsulation of multiple antibiotics within nanoparticles significantly reduces the likelihood of a single bacterium developing resistance to multiple drugs, as this would necessitate the occurrence of multiple gene mutations [250]. Consequently, the incorporation of multiple drugs within a single nanoparticle has the potential to enhance the potency of the mixture, thereby improving its antimicrobial efficacy and the likelihood of overcoming existing resistance mechanisms in microorganisms [251].

Nanoparticles can be metallic particles or lipid-based liposomes. For metallic nanoparticles, gold nanoparticles could impede the formation of biofilms by new bacteria on medical devices and prevent the adhesion of new bacterial cells to existing biofilms [252]. Consequently, gold nanoparticles can be employed in conjunction with antibiotics such as ampicillin and streptomycin, against a broad spectrum of drug-resistant organisms, including *P. aeruginosa* [253]. On the other hand, liposome nanoparticles can enable better incorporation into the bacterial plasma membrane and a one-time release of the contained drug [246], resulting in a faster and better drug delivery. Binding liposomes to polymyxin B can reduce antibiotic resistance in *P. aeruginosa*, leading to rapid lipidation of *P. aeruginosa* membranes [246].

The application of nanoparticles has been shown to enhance the delivery and efficacy of drugs through multiple mechanisms. Nanoparticles can improve the solubility and stability of drugs that are poorly water-soluble, thereby increasing their bioavailability and therapeutic potential [254]. Through the modification of their surface with polymers such as polyethylene glycol (PEG), nanoparticles can reduce opsonization and extend their circulation time in the bloodstream, resulting in greater accumulation at infection sites [255]. The functionalization of nanoparticles with ligands, such as antibodies or peptides, enables a targeted delivery of nanoparticles to *P. aeruginosa* with minimized off-target effects [256]. Furthermore, nanoparticles can overcome physiological barriers, such as the blood–brain barrier or epithelial layers, to deliver drugs to otherwise inaccessible infection sites [257].

## 4. Future Directions and Research Needs

### 4.1. Need for More Robust Clinical Trials for Alternative Therapies

In the context of mounting antibiotic resistance, it is imperative to conduct more comprehensive clinical trials to evaluate alternative therapeutic modalities, including phage therapy, CRISPR-Cas systems and AMPs. These studies should concentrate on ascertaining the safety and efficacy of these innovative therapies in diverse patient populations, particularly those with chronic infections [258,259,260]. Collaboration between academic institutions, pharmaceutical companies and healthcare providers is essential to facilitate these trials.

### 4.2. Potential Regulatory Challenges in Developing and Approving New Treatments

The process of developing and obtaining regulatory approval for new therapeutic interventions for antibiotic-resistant infections is often characterized by numerous and substantial regulatory challenges. Engagement with regulators at an early stage of the development process has been demonstrated to facilitate the approval process for novel therapies [261]. Furthermore, policymakers must consider the creation of frameworks that encourage the rapid evaluation of novel antimicrobials and alternative therapies, while balancing safety and efficacy [232].

In summary, concerted efforts are required to enhance our understanding of the mechanisms underlying *P. aeruginosa* resistance, with the objective of developing effective treatments. By prioritizing these research areas, we can establish the foundation for innovative strategies that will contribute to a significant reduction in the burden of antibiotic-resistant infections.

## Figures and Tables

**Figure 1 microorganisms-13-00913-f001:**
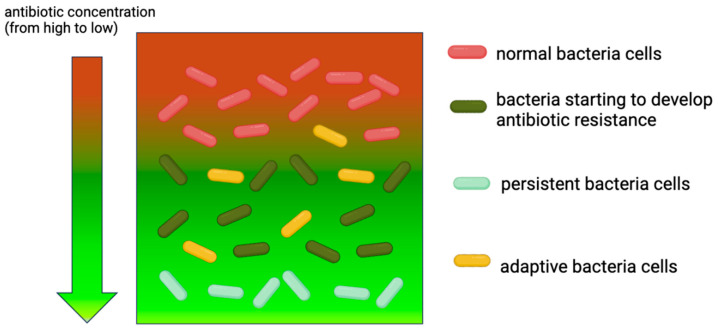
**Persistent cell and adaptive cell mutation mechanisms.** By slowly penetrating into the biofilm matrix (green), antibiotics (red) gradually add selective pressure on biofilm bacteria. Some bacteria cells inside the biofilm develop adaptive mutation (yellow) in a harsh environment. Antibiotic existence alters the microenvironment of biofilm, which leads to a bacteria mutation helping to export antibiotics (dark green). In the meantime, MDR persister cells are formed (light green).

**Figure 2 microorganisms-13-00913-f002:**
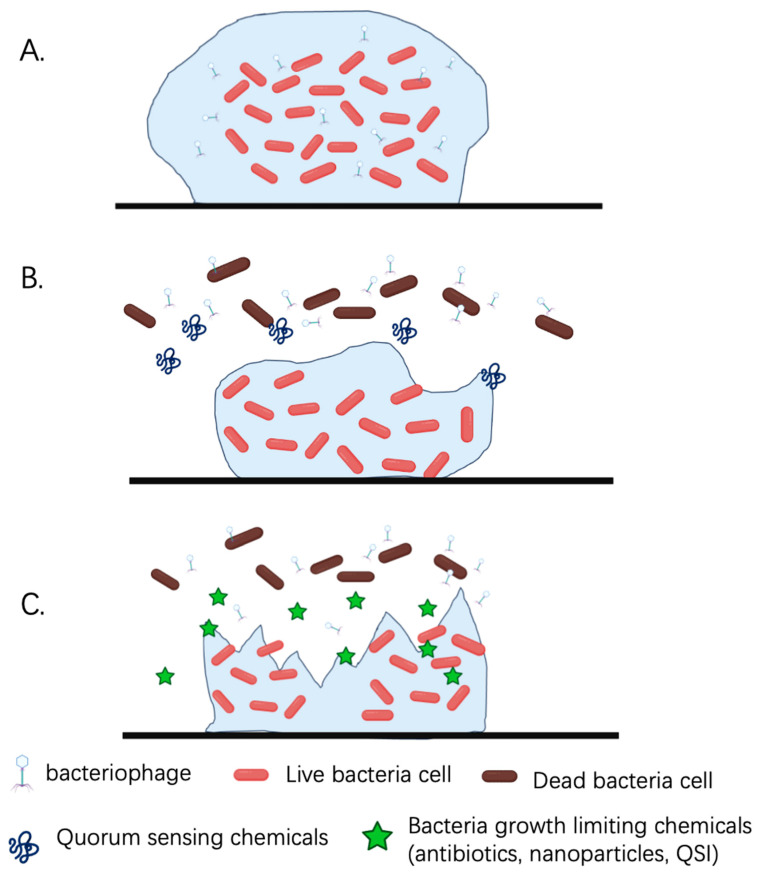
**Mechanisms of phage therapy and its combinational methods against *P. aeruginosa* biofilms.** (**A**) Phages can penetrate into the biofilm effectively and eliminate resident bacteria. (**B**) Phages inhibit biofilm formation by inhibiting QS and reducing cellular communication. (**C**) Combined use of phages and other bacteria-growth-limiting chemicals (antibiotics, nanoparticles, QSI, etc.).

**Table 1 microorganisms-13-00913-t001:** Common genes and plasmids associated with AMR in *P. aeruginosa*, listed as gene and followed by plasmids. The order of gene was set by the frequency of antibiotic mutation and same method applied to the order of plasmids listed.

Categories	Name	Mechanism	Function	Reference
Metallo-β-lactamases	*blaVIM*, *blaIMP*, and *blaNDM*	encode metallo-β-lactamases (MBLs),	confer resistance to carbapenems	[42]
Serine carbapenemases	*blaOXA*	encodes oxacillinases	confer resistance to carbapenems	[43]
Chromosomal β-Lactamase	*ampC*	overexpression of ampC, a chromosomal β-lactamase	hydrolyzes carbapenems, particularly when combined with oprD mutations.	[44]
pmrAB (PhoPQ) Two-Component System	*pmrAB*, *phoPQ*	lead to LPS modifications	decreasing colistin binding	[45]
Lipid A Biosynthesis Genes	*lpxA*, *lpxC*, and *lpxD*	lead to the complete loss of LPS production	resulting in colistin resistance through the absence of the drug’s primary target	[46]
Efflux Pump Genes	*oprD*, *mexX*, *mexY*	encode multidrug efflux pumps	expel a wide range of antibiotics	[47]
Aminoglycoside Resistance Genes	*aac*, *aph*, and *aad*	encode enzymes that modify aminoglycosides, rendering them ineffective	confer resistance to aminoglycoside	[48]
Quinolone Resistance Genes	*qnr*	encodes proteins that protect DNA gyrase and topoisomerase IV from quinolone inhibition,	confer resistance to quinolone	[49]
Key Plasmids Associated with AMR	pMOS94	carries the *blaVIM-1* gene within a class 1 integron (In70)	associated with the dissemination of metallo-β-lactamase genes among *Pseudomonas* species	[42]
	pBT2436 and pBT2101	carry extensive arrays of antibiotic resistance genes, including *blaOXA-10, VEB-1,* and *aadA.*	confer resistance to carbapenems, aminoglycoside and β-lactam	[50]
	pUM505	encodes a mobile genetic element (*Mpe*) that confers resistance to heavy metals (e.g., chromate and mercury) and enhances virulence.		[51]
	pAER57	associated with the spread of *blaVIM-2*	confer resistance to β-lactam	[42]
	pMKPA34-1	carries resistance genes such as *mexCD-oprJ* and is associated with multidrug resistance	confer resistance to multiple antibiotics by inducing antibiotic efflux.	[50]

**Table 2 microorganisms-13-00913-t002:** Key AMPs with activity against *P. aeruginosa*. The antimicrobial peptides in the table were listed in alphabetical order based on their names, with no explicit ranking or prioritization.

Peptide Name	Origin	Characteristics	Applications/Effects on *P. aeruginosa*	Reference
**Colistin (Polymyxin E)**	*Bacillus polymyxa*	Cationic, cyclic lipopeptide; targets lipopolysaccharides (LPS) in Gram-negative bacteria	Last-resort treatment for MDR/XDR *P. aeruginosa*; disrupts bacterial membrane integrity, leading to cell lysis.	[50]
**LL-37**	Human cathelicidin	Cationic, α-helical peptide; part of the innate immune system	Broad-spectrum activity; disrupts *P. aeruginosa* biofilms and enhances immune cell recruitment.	[167]
**Melittin**	Honeybee venom	Cationic, α-helical peptide; highly amphipathic	Disrupts bacterial membranes; effective against *P. aeruginosa* biofilms and planktonic cells.	[168]
**Pexiganan (MSI-78)**	Synthetic analog of magainin	Cationic, α-helical peptide; derived from frog skin peptides	Targets *P. aeruginosa* membranes; used in topical treatments for wound infections.	[169]
**Polymyxin B**	*Bacillus polymyxa*	Cationic, cyclic lipopeptide; similar to colistin	Used against MDR/XDR *P. aeruginosa*; disrupts LPS and membrane integrity.	[170]
**Lactoferrin**	Mammalian secretions (e.g., milk, tears)	Iron-binding glycoprotein; cationic and multifunctional	Inhibits *P. aeruginosa* biofilm formation; enhances the activity of other antibiotics.	[171]
**Cecropin A**	Silk moth (*Hyalophora cecropia*)	Cationic, α-helical peptide; broad-spectrum activity	Disrupts *P. aeruginosa* membranes; effective against planktonic and biofilm-associated cells.	[172]
**Defensins (e.g., HBD-1, HBD-2)**	Human epithelial cells	Cationic, β-sheet peptides; part of the innate immune system	Disrupts *P. aeruginosa* membranes and biofilms; enhances immune responses.	[173]
**Thanatin**	Spined soldier bug (*Podisus maculiventris*)	Cationic, β-hairpin peptide; broad-spectrum activity	Targets *P. aeruginosa* membranes and inhibits outer membrane protein assembly.	[174]
**Epidermin**	*Staphylococcus epidermidis*	Lantibiotic; post-translationally modified peptide	Disrupts *P. aeruginosa* membranes; effective against biofilms and planktonic cells.	[175]
**Plectasin**	Fungus (*Pseudoplectania nigrella*)	Defensin-like peptide; cationic and stable	Disrupts *P. aeruginosa* membranes; effective against MDR strains.	[176]

**Table 3 microorganisms-13-00913-t003:** Examples of QSIs currently under study and their efficacy in clinical models. The QSIs are organized primarily by the source/origin of the compounds, with a secondary focus on natural versus synthetic products. AHL: acyl homoserine lactones; HSL: homoserine lactone.

Name	Source	Chemical Property	Inhibition Mechanism	Effect	References
**Ajoene**	Garlic (*Allium sativum* L.)	Natural product	Down-regulation of QS genes (*lasA*, *chiC* and *rhlAB*)	Inhibition of virulence and biofilm formation	[187,188]
**Iberin**	Horseradish extracts (*Armoracia rusticana*)	Natural product	Antagonists of LasIR and RhlI/R	Inhibition of virulence and biofilm formation	[189]
**Sulforaphane**	Broccoli extracts (*Brassica oleracea*)	Natural product	Antagonist of LasR	Inhibition of virulence and biofilm formation	[190]
**Phenolics**	Ginger extract (*Curcuma longa*)	Natural product	Down-regulation of LasI by binding of the compound’s long acyl chain to LasR	Inhibition of virulence and biofilm formation	[191]
**Caffeine**	Fenugreek seeds extract (*Trigonella foenum-graecum* L.)	Natural product	Inhibit AHL production	Inhibition of virulence and biofilm formation	[192]
**Flavan-3-ol catechin**	Malagasy plant (*Combretum albiflorum*)	Natural product	Reduced signal perception of RhlR	Inhibition of virulence and biofilm formation	[193]
* **Kalanchoe** * **leaves extract**	*Kalanchoe blossfeldiana*	Natural product	Interference with AHL production	Inhibition of virulence and biofilm formation	[194]
**Clove oil**	*Syzygium aromaticum*	Natural product	Inhibition of QS-mediated biofilm formation and disruption of already formed *P. aeruginosa*	Inhibition of virulence and biofilm formation	[195]
**Juglone**	Green part of *Juglans regia*	Natural product	bind to the PqsR active site	Inhibition of QS-mediated biofilms and reduction in virulence factor production	[196]
**Quercetin (flavanols)**	Apples, grapes, onions, tomatoes, etc.	Natural product	*Reduced expression of the QS genes lasI, lasR, rhlI and rhlR*	Significantly reduces biofilm formation	[197]
**AHL-lactonase**	*Bacillus* spp. gene *aiiA*	Natural product	Break down AHL	Prevents biofilm formation and reduces virulence factors of many bacteria	[198]
**Paraoxonase**	Human epithelial cells and serum from mammals such as rats, goats, cows and horses	Natural product	Inhibition of AHL-mediated QS in *P. aeruginosa*	Inhibition of virulence and biofilm formation	[199]
**Vanillin**	primary phenolic aldehyde of vanilla bean	Natural product	inhibited pqs expression and its associated phenotypes production	Inhibition of biofilm growth and reducing virulence	[200]
**MAb RS2-1G9**		antibody	analog of the AHL acyl chain	Against the production of chlorpyrifosin in *P. aeruginosa*	[201]
**MAb XYD-11G2**		antibody	Hydrolysed 3-oxo-C12-HSL	Inhibition of pyocyanin production by *P. aeruginosa*	[201]
**Macrolides, including azithromycin and erythromycin**		Synthetic product	Reducing transcription of *lasI* and *rhlI* simultaneously reduce the concentration of 3-oxo-C12-HSL and C4-HSL	Reducing the production of group induction-dependent virulence factors.	[202,203]
**Thiazolidinedione (TZD) and its derivatives**		Synthetic product	-	A 70 percent reduction in biomass in biofilm	[204]
**Furanone compounds**		Synthetic product	-	Inhibition of QS and reducing bacteria growth	[205]
**Cyclohexanone analog of HSL**		Synthetic product	-	Effective antagonism of QS-mediated activity, including biofilm formation	[206]
**Engineered variant of hyper-thermostable lactonase** * **Sso** * **Pox**	*Sulfolobus solfataricus*	Synthetic molecule	Degradation of the lactone ring of 3-oxo-C12 AHL and enhancement of catalytic efficiency	Reducing the severity of pneumonia caused by *P. aeruginosa* infection	[207]

**Table 4 microorganisms-13-00913-t004:** Mechanisms of pathogen-centered CRISPR therapies for *K. pneumoniae*, *C. difficile* and *S. aureus.* Only the most common bacterial species were selected and the order of the bacteria was listed in alphabetical order.

Bacteria	Gene	Mechanism	Reference
*C. difficile*	*RNase Y*	Using bacteriophage ϕCD24-2 expressing bacterial genome-targeting crRNAs/chromosomal DNA degradation	[224]
*K. pneumoniae*	*ramR*, *tetA*, *mgrB*	pSGKP-spe and pBECKP-spe plasmids engineered with the CRISPR-Cas9 system	[226]
*S. aureus*	*nuc*, *esxA*	Using Mild Phage ϕSaBov as chromosomal DNA degradation delivery	[222]

## Data Availability

No new data were created or analyzed in this study. Data sharing is not applicable to this article.

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
