# Peer review of "Antibiotic-Resistant Pseudomonas aeruginosa: Current Challenges and Emerging Alternative Therapies"

_microorganisms, 2025, doi:10.3390/microorganisms13040913_

Round 1

Reviewer 1 Report

Comments and Suggestions for Authors

The review article submitted by Hu & Chua discusses alternative methodologies for controlling the opportunistic pathogen Pseudomonas aeruginosa when resistant to antimicrobials, the mechanisms involved in the development of antimicrobial resistance (AMR), and emerging strategies for its control.

This is a highly relevant and timely topic, as AMR represents a severe global public health threat, and P. aeruginosa is a major etiological agent of infections in both humans and animals. In my opinion, the review is well-developed and well-written, requiring only minor revisions.

Revisions

  • In the Introduction, I suggest that the authors include information regarding the significance of P. aeruginosa as an etiological agent of infections, providing data on its prevalence and impact. Additionally, it would be valuable to incorporate data illustrating the importance of AMR as a global public health concern and, if available, specific data on the impact of AMR in P. aeruginosa.
  • L35: Provide a reference.
  • L48: Provide a reference.
  • In Section 2.1.1, include a basic explanation of the mechanism of efflux pumps.
  • L98: Use italics.
  • L140: "Genes" should be in the plural form, as multiple genes are involved.
  • In Section 2.2.1, it is important to mention the key genes and plasmids associated with AMR due to mobile genetic elements in this microorganism.
  • L114: Provide a reference.
  • L174: Use the full term instead of an abbreviation.
  • L175–176: Define the concepts of MDR (multidrug resistance) and XDR (extensively drug-resistant).
  • L201: Use italics.
  • L216: "Microbiota" instead of "microflora."
  • L223: Provide a reference.
  • L224: Use italics. Double-check the correct formatting of all scientific names throughout the manuscript.
  • L290: Verify the correct use of the reference.
  • In Section 3.2, I recommend including a descriptive table listing key peptides, their origins, characteristics, and applications. The section is currently somewhat superficial and requires more detailed descriptions of the primary peptides that have been studied for their effects on P. aeruginosa. This table should be similar to Table 1.
  • In Table 1, use italics for gene names and ensure this formatting is consistently applied throughout the manuscript.
  • Table 3: "K. pneumoniae"
  • L398: Use "Rodrigues and colleagues"
  • Ensure that all mandatory sections are included according to the journal’s template and author guidelines.

Author Response

The review article submitted by Hu & Chua discusses alternative methodologies for controlling the opportunistic pathogen Pseudomonas aeruginosa when resistant to antimicrobials, the mechanisms involved in the development of antimicrobial resistance (AMR), and emerging strategies for its control.

This is a highly relevant and timely topic, as AMR represents a severe global public health threat, and P. aeruginosa is a major etiological agent of infections in both humans and animals. In my opinion, the review is well-developed and well-written, requiring only minor revisions.

Revisions

In the Introduction, I suggest that the authors include information regarding the significance of P. aeruginosa as an etiological agent of infections, providing data on its prevalence and impact. Additionally, it would be valuable to incorporate data illustrating the importance of AMR as a global public health concern and, if available, specific data on the impact of AMR in P. aeruginosa.

Response:

Thank you for your excellent suggestion. As suggested, we had inserted further information on the significance of P. aeruginosa in infections and AMR, in Pages 3-4, Lines 44-52, as follows:

It is a leading cause of nosocomial infections, accounting for approximately 7.1%–7.3% of all HAIs [1], with higher prevalence in intensive care units (ICUs), where it is responsible for up to 16.2% of infections. It has been observed that P. aeruginosa is associated with a number of adverse outcomes, including ventilator-associated pneumonia (VAP), surgical site infections, urinary tract infections (UTIs), and bloodstream infections (BSIs) [1]. In severe cases, such as VAP and BSIs, the mortality rate can range from 32% to 58.8% [1]. In cystic fibrosis patients, chronic P. aeruginosa infections are a major contributor to morbidity and mortality, with prevalence rates as high as 49.6% in some populations [1]. The impact of P. aeruginosa is further exacerbated by its intrinsic and acquired antimicrobial resistance (AMR), making it a critical public health concern. The prevalence of multidrug-resistant (MDR) and extensively drug-resistant (XDR) strains has escalated, with resistance rates to carbapenems exceeding 30% in certain regions [2]. This resistance complicates treatment, resulting in prolonged hospital stays, escalated healthcare costs, and increased mortality. The ability of the pathogen to form biofilms and to produce virulence factors such as exotoxins and quorum-sensing molecules serves to enhance its persistence and pathogenicity, thus rendering it a formidable challenge in clinical settings.

L35: Provide a reference.

Response:

Thank you for your comment. We have provided the reference in Page 5, Lines 99-101, as follows: Furthermore, inadequacy of the drug development pipeline is also contributing to the failure to generate effective treatments comparing with traditional antibiotic treatment [3].

L48: Provide a reference.

Response:

Thank you for your comment. We have provided the reference in Page 5, Lines 102, as follows: efficacy of treatments [6].

In Section 2.1.1, include a basic explanation of the mechanism of efflux pumps.

Response:

Thank you for your comment. We have included a basic explanation of the mechanism of efflux pumps in Page6, Lines 125-131, as follows:

Efflux pumps can pump specific substrates, including various classes of antibiotics (e.g., fluoroquinolones, beta-lactams, tetracyclines) and other toxic molecules across the cell membrane and out of the bacterial cells [4], with the use of ATP or proton motive force [5]. The expulsion of these substrates from the cell results in a reduction of their intracellular concentration and prevents them from reaching their target sites, including ribosomes, DNA gyrase, and cell wall synthesis machinery [4].

L98: Use italics.

Response:

Thank you for your comment. We have used italics for the bacterial species in Page 8, Lines 179-180 as follows:

Acquired resistance is another approach by which P. aeruginosa acquires antibiotic resistance, including horizontal gene transfer and accumulation of genetic mutations.

L140: "Genes" should be in the plural form, as multiple genes are involved.

Response:

Thank you for your comment. We have changed into plural form in Page 11, Lines 212, as follows:

148 genes sequences.

In Section 2.2.1, it is important to mention the key genes and plasmids associated with AMR due to mobile genetic elements in this microorganism.

Response:

Thank you for the good suggestion. We have mentioned the key genes and plasmids associated with AMR in Page 9, Lines 199, as follows:

Table 1 Common genes and plasmids associated with AMR in P. aeruginosa

Categories

Name

Mechanism

Function

References

Metallo-β-lactamases

blaVIM, blaIMP, and blaNDM

encode metallo-β-lactamases (MBLs),

confer resistance to carbapenems

[6]

Serine carbapenemases

blaOXA

encodes oxacillinases

confer resistance to carbapenems

[7]

Chromosomal β-Lactamase

ampC

overexpression of ampC, a chromosomal β-lactamase

hydrolyzes carbapenems, particularly when combined with oprD mutations.

[8]

pmrAB (PhoPQ) Two-Component System

pmrABphoPQ

lead to LPS modifications

decreasing colistin binding

[9]

Lipid A Biosynthesis Genes

lpxA, lpxC, and lpxD

lead to the complete loss of LPS production

resulting in colistin resistance through the absence of the drug's primary target

[10]

Efflux Pump Genes

oprD, mexX, mexY

encode multidrug efflux pumps

expel a wide range of antibiotics

[11]

Aminoglycoside Resistance Genes

aac, aph, and aad

encode enzymes that modify aminoglycosides, rendering them ineffective

confer resistance to aminoglycoside

[12]

Quinolone Resistance Genes

qnr

encodes proteins that protect DNA gyrase and topoisomerase IV from quinolone inhibition,

confer resistance to quinolone

[13]

Key Plasmids Associated with AMR

pMOS94

carries the blaVIM-1 gene within a class 1 integron (In70)

associated with the dissemination of metallo-β-lactamase genes among Pseudomonas species

[6]

pBT2436 and pBT2101

carry extensive arrays of antibiotic resistance genes, including blaOXA-10, VEB-1, and aadA.

confer resistance to carbapenems, aminoglycoside and β-lactam

[14]

pUM505

encodes a mobile genetic element (Mpe) that confers resistance to heavy metals (e.g., chromate and mercury) and enhances virulence.

[15]

pAER57

associated with the spread of blaVIM-2 

confer resistance to β-lactam

[6]

pMKPA34-1

carries resistance genes such as mexCD-oprJ and is associated with multidrug resistance

confer resistance to multiple antibiotics by inducing antibiotic efflux.

[14]

L114: Provide a reference.

Response:

Thank you for your comment. We have provided the reference in Page 14, Lines 274-275, as follows:

C-di-GMP, a small intracellular molecule that is the second messenger of bacterial metabolic pathway, plays a pivotal role in regulating the growth of biofilm [54].

L174: Use the full term instead of an abbreviation.

Response:

Thank you for your comment. We have provided the reference in Page 14, Lines 289-292, as follows: The toxin-antitoxin system usually consists of two gene parts in a manipulator, one responsible for encoding a stable toxin that disrupts life processes such as mRNA translation, and the other responsible for encoding an unstable toxin such as RNA or protein that prevents toxicity[16].

L175–176: Define the concepts of MDR (multidrug resistance) and XDR (extensively drug-resistant).

Response:

Thank you for your comment. We have provided the definition in Page 4, Lines 73-76, as follows:

MDR refers to resistance to multiple antimicrobial agents within three or more classes of antibiotics, while XDR refers to resistance to almost all antimicrobial agents across multiple antibiotic classes, with susceptibility to only one or two categories of antibiotics [17].

L201: Use italics.

Response:

Thank you for your comment. We have used italics in Page 17, Lines 379 as follows: inhibit P. aeruginosa biofilm.

L216: "Microbiota" instead of "microflora."

Response:

Thank you for your comment. We have made the change in Page 19, Lines 387 as follows: microbiota in the human body.

L223: Provide a reference.

Response:

Thank you for your comment. We have provided the reference in Page 19, Lines 395-397 as follows:

As the bacteria are killed, the titer of the phage decreases until the phage is completely eliminated from the patient's body [18].

L224: Use italics. Double-check the correct formatting of all scientific names throughout the manuscript.

Response:

Thank you for your comment. We have used italics in Page 19, Lines 399 as follows:

eradicating P. aeruginosa strains

L290: Verify the correct use of the reference.

Response:

Thank you for your comment. We have edited the reference in Page 21, Lines 466-467 as follows:

The potential of phages (pp1131) in the treatment of P. aeruginosa endocarditis has also been demonstrated in a mouse model.[19].

In Section 3.2, I recommend including a descriptive table listing key peptides, their origins, characteristics, and applications. The section is currently somewhat superficial and requires more detailed descriptions of the primary peptides that have been studied for their effects on P. aeruginosa. This table should be similar to Table 1.

Response:

Thank you for your great suggestion. We have inserted the table in Page 23-25, Lines 516 as follows:

Table2. Key AMPs with Activity Against P. aeruginosa

Peptide Name

Origin

Characteristics

Applications/Effects on P. aeruginosa

References

Colistin (Polymyxin E)

Bacillus polymyxa

Cationic, cyclic lipopeptide; targets lipopolysaccharides (LPS) in Gram-negative bacteria

Last-resort treatment for MDR/XDR P. aeruginosa; disrupts bacterial membrane integrity, leading to cell lysis.

[14]

LL-37

Human cathelicidin

Cationic, α-helical peptide; part of the innate immune system

Broad-spectrum activity; disrupts P. aeruginosa biofilms and enhances immune cell recruitment.

[20]

Melittin

Honeybee venom

Cationic, α-helical peptide; highly amphipathic

Disrupts bacterial membranes; effective against P. aeruginosa biofilms and planktonic cells.

[21]

Pexiganan (MSI-78)

Synthetic analog of magainin

Cationic, α-helical peptide; derived from frog skin peptides

Targets P. aeruginosa membranes; used in topical treatments for wound infections.

[22]

Polymyxin B

Bacillus polymyxa

Cationic, cyclic lipopeptide; similar to colistin

Used against MDR/XDR P. aeruginosa; disrupts LPS and membrane integrity.

[23]

Lactoferrin

Mammalian secretions (e.g., milk, tears)

Iron-binding glycoprotein; cationic and multifunctional

Inhibits P. aeruginosa biofilm formation; enhances the activity of other antibiotics.

[24]

Cecropin A

Silk moth (Hyalophora cecropia)

Cationic, α-helical peptide; broad-spectrum activity

Disrupts P. aeruginosa membranes; effective against planktonic and biofilm-associated cells.

[25]

Defensins (e.g., HBD-1, HBD-2)

Human epithelial cells

Cationic, β-sheet peptides; part of the innate immune system

Disrupts P. aeruginosa membranes and biofilms; enhances immune responses.

[26]

Thanatin

Spined soldier bug (Podisus maculiventris)

Cationic, β-hairpin peptide; broad-spectrum activity

Targets P. aeruginosa membranes and inhibits outer membrane protein assembly.

[27]

Epidermin

Staphylococcus epidermidis

Lantibiotic; post-translationally modified peptide

Disrupts P. aeruginosa membranes; effective against biofilms and planktonic cells.

[28]

Plectasin

Fungus (Pseudoplectania nigrella)

Defensin-like peptide; cationic and stable

Disrupts P. aeruginosa membranes; effective against MDR strains.

[29]

In Table 1, use italics for gene names and ensure this formatting is consistently applied throughout the manuscript.

Response:

Thank you for your comment. We have edited the gene names in Table 2 (line 516) as follows:

lasA, chiC and rhlAB  & lasI, lasR, rhlI and rhlR

Table 3: "K. pneumoniae"

Response:

Thank you for your comment. We have edited the bacteria name in the new Table 4, Lines 606-607 as follows:

Table 4. Mechanism of pathogen-centered CRISPR therapy of S. aureus, C. difficile and K. pneumoniae.

L398: Use "Rodrigues and colleagues"

Response:

Thank you for your comment. We have provided the reference in Page 35, Lines 630-634 as follows:

Next, Rodrigues and colleagues [30] successfully modified the splice plasmid pPD1 using an intact, constitutively expressed CRISPR-Cas9 system, thus enabling the excision of erythromycin (ermB) and tetracycline (tetM) resistance genes present within the plasmid, resulting in a notable reduction of antibiotic resistance both in vivo and in vitro.

Reviewer 2 Report

Comments and Suggestions for Authors

The authors have conducted an interesting review on the potential treatments for Pseudomonas aeruginosa infections, analyzing the impact of antibiotic therapies and evaluating possible therapeutic alternatives. Overall, this is a high-quality manuscript that explores one of the emerging challenges of recent years—multidrug-resistant infections—due to the increasing incidence of bacterial resistance. Below are my comments:  

- Line 21: In the abstract, I suggest spelling out the acronym in full.  
- Line 41: It would be beneficial to further elaborate on urinary tract infections (UTIs), which are among the most common infections associated with Pseudomonas, such as catheter-associated UTIs or those occurring in patients with urinary tract anomalies.  
- Line 176: What are the current first-line antibiotics used to treat multidrug-resistant Pseudomonas infections? It would be useful for the reader to explore this topic further, possibly in a dedicated subsection.  
- Regarding the potential treatments for multidrug-resistant infections, I recommend expanding the discussion on prevention strategies. For instance, Pseudomonas urinary infections in the pediatric population have been shown to respond optimally to short-course antibiotic therapy (see 10.3390/children9111647), achieving therapeutic efficacy comparable to standard regimens while reducing the incidence of antibiotic resistance and reinfection. This concept is well applicable to UTIs but is more challenging to implement in chronic patients, such as those with cystic fibrosis.  
- Table 1: Please add a legend specifying the abbreviations used in the text.  
- Minor improvements to the English translation of the manuscript are needed.  

I thank the authors for their excellent work and look forward to reviewing the revised version of the manuscript.

Comments on the Quality of English Language

Minor improvements to the English translation of the manuscript are needed. 

Author Response

The authors have conducted an interesting review on the potential treatments for Pseudomonas aeruginosa infections, analyzing the impact of antibiotic therapies and evaluating possible therapeutic alternatives. Overall, this is a high-quality manuscript that explores one of the emerging challenges of recent years—multidrug-resistant infections—due to the increasing incidence of bacterial resistance. Below are my comments:  

Line 21: In the abstract, I suggest spelling out the acronym in full.  
Response:

Thank you for your comment. We have spelt out the acronym in full in Page 2, Line 22-23, as follows:

With traditional antibiotic therapy rendered ineffective against Pseudomonas aeruginosa infections, we explore alternative therapies that have shown promise, including antimicrobial peptides, nanoparticles and quorum sensing inhibitors.

Line 41: It would be beneficial to further elaborate on urinary tract infections (UTIs), which are among the most common infections associated with Pseudomonas, such as catheter-associated UTIs or those occurring in patients with urinary tract anomalies.  
Response:

Thank you for your good suggestion, so we have added the UTI part in Page 3, Line 55-61, as follows:

Moreover, urinary tract infections (UTIs) caused by P. aeruginosa pose a significant clinical challenge, particularly within healthcare settings, due to their association with catheter use, urinary tract abnormalities, and immunocompromised states [31]. The intrinsic and acquired resistance mechanisms of pathogens, including biofilm formation and the capacity to invade bladder epithelial cells, contribute to the complexity of treatment regimens for these infections. P. aeruginosa is a major cause of catheter-associated urinary tract infections (CAUTI), and biofilms on catheter surfaces complicate eradication [31].

Line 176: What are the current first-line antibiotics used to treat multidrug-resistant Pseudomonas infections? It would be useful for the reader to explore this topic further, possibly in a dedicated subsection.  
- Regarding the potential treatments for multidrug-resistant infections, I recommend expanding the discussion on prevention strategies. For instance, Pseudomonas urinary infections in the pediatric population have been shown to respond optimally to short-course antibiotic therapy (see 10.3390/children9111647), achieving therapeutic efficacy comparable to standard regimens while reducing the incidence of antibiotic resistance and reinfection. This concept is well applicable to UTIs but is more challenging to implement in chronic patients, such as those with cystic fibrosis.  
2.4 current first-line antibiotics used to treat MDR Pseudomonas infections

Response:

Thank you for your great suggestions. We have added the current first-line antibiotics used to treat multidrug-resistant Pseudomonas infections in Page 15-16, Line 313-344, as follows:

2.4 Current first-line antibiotics used to treat MDR Pseudomonas infections

The treatment of MDR P. aeruginosa infections poses significant challenges due to its intrinsic and acquired resistance mechanisms, so several new antibiotic combinatorial treatments have emerged as first-line options for treating such infections. Cefazolin-tazobactam, a novel combination of cephalosporins and β-lactamase inhibitors, has been shown to be highly effective against MDR and carbapenem-resistant P. aeruginosa [32], particularly in complicated intra-abdominal infections [32], urinary tract infections [33] and hospital-acquired pneumonia [34]. Cefozoxime-avibactam, another beta-lactam-beta-lactamase inhibitor combination, was effective against broad-spectrum beta-lactamase-producing and carbapenem-resistant strains, with a susceptibility rate of over 95% against P. aeruginosa [35]. A glycosidophilic cephalosporin, cefiderocol, exploits the bacterial iron transport system to overcome resistance mechanisms such as exocytosis pumps and pore-protein mutations, maintaining its activity against 88.7% of P. aeruginosa isolates [36]. Finally, polymyxin antibiotics, such as colistin, remain the last option for the treatment of severely and XDR infections [37], but their use is limited by significant toxicity and emerging colistin resistance [38]. Consequently, there is an urgent need for effective therapies against drug-resistant P. aeruginosa strains that are less toxic and have a broader range of action than conventional antibiotics.

Prevention strategies also shed light as a potential treatment against P. aeruginosa infections. In the treatment of uncomplicated UTIs in children, short courses of antibiotic therapy (5-7 days) have been shown to be comparable in efficacy to the standard 10-day to 14-day course of therapy [39]. These findings are particularly salient in light of the concomitant reduction in antibiotic exposure, development of resistance, and adverse effects [39]. A meticulous examination of randomized controlled trials reveals no statistically significant disparity in recurrence rates between short courses and standard therapy, thereby substantiating the efficacy of this approach in uncomplicated UTIs caused by P. aeruginosa [40]. However, in chronic diseases such as cystic fibrosis, where infections are often recurrent and associated with biofilms [41], short courses of therapy are less effective and require extended or suppressive antibiotic regimens [42].

  

Table 1: Please add a legend specifying the abbreviations used in the text.  
Response:

Thank you for your comment. We have provided the reference in Page 27, Lines 554, as follows: AHL: acyl homoserine lactones; HSL: homoserine lactone

Minor improvements to the English translation of the manuscript are needed.  

Response:

Thank you for your comment. We have asked an English reader to check our manuscript and improve the sentences.

Reviewer 3 Report

Comments and Suggestions for Authors

Minor comments:

How do biofilms contribute to the antibiotic resistance of Pseudomonas aeruginosa, and what are the main components of the extracellular polymeric substance (EPS) that protect bacteria? (Lines 128–137)

What is the role of efflux pumps in intrinsic resistance, and how do the different efflux pump families contribute to multidrug resistance in P. aeruginosa? (Lines 71–85)

How does horizontal gene transfer facilitate the acquisition of antibiotic resistance genes in P. aeruginosa, and what are the main genetic elements involved? (Lines 100–105)

What are the key mutations that contribute to adaptive resistance in P. aeruginosa, particularly in response to colistin and carbapenems? (Lines 107–120)

How does the quorum sensing (QS) system regulate biofilm formation, and what are the potential therapeutic targets within QS pathways? (Lines 147–152)

What advantages does phage therapy offer over conventional antibiotics in treating antibiotic-resistant P. aeruginosa, and what are its primary limitations? (Lines 211–261)

How do antimicrobial peptides (AMPs) target P. aeruginosa, and what challenges exist in their clinical application? (Lines 305–334)

What are the mechanisms by which quorum sensing inhibitors (QSIs) disrupt bacterial communication, and how can P. aeruginosa develop resistance to QSIs? (Lines 337–370)

How can CRISPR-Cas systems be leveraged to combat antibiotic resistance in P. aeruginosa, and what challenges exist in their clinical application? (Lines 371–421)

What potential do nanoparticles have in overcoming drug resistance in P. aeruginosa, and how do they enhance drug delivery and efficacy? (Lines 422–452)

Author Response

Minor comments:

How do biofilms contribute to the antibiotic resistance of Pseudomonas aeruginosa, and what are the main components of the extracellular polymeric substance (EPS) that protect bacteria? (Lines 128–137)

Response:

Thank you for your constructive comment. We have added EPS part in Page 12 Lines 228-237, as follows:

The role of biofilms in antibiotic resistance is attributable to their physicochemical properties. Polysaccharides, including alginate, Psl and Pel, provide structural integrity and adhesion; extracellular DNA (eDNA) stabilises the matrix and neutralises antibiotics; proteins promote cohesion; and lipids increase hydrophobicity [43, 44]. The viscous nature of the EPS reduces the penetration and utilization efficiency of antibiotics [45], so biofilms are more resistant to antibiotics than planktonic cells [46]. Moreover, the EPS acts as a physical barrier which harbors metabolically inactive persister cells, promotes horizontal gene transfer of antibiotic resistance genes, and protects bacterial cells from predation and detection of bacterial-secreted metabolites by predators [47-49].

What is the role of efflux pumps in intrinsic resistance, and how do the different efflux pump families contribute to multidrug resistance in P. aeruginosa? (Lines 71–85)

Response:

Thank you for your useful suggestions. We have added the role of efflux pumps in Page 6, Line 123-136, as follows:

Efflux pumps play crucial roles in developing the intrinsic antibiotic resistance in P. aeruginosa. Their ability to export the antibiotics provide P. aeruginosa with extra survival chance under antibiotic choice stress. Efflux pumps can pump specific substrates, including various classes of antibiotics (e.g., fluoroquinolones, beta-lactams, tetracyclines) and other toxic molecules across the cell membrane and out of the bacterial cells [4], with the use of ATP or proton motive force [5]. The expulsion of these substrates from the cell results in a reduction of their intracellular concentration and prevents them from reaching their target sites, including ribosomes, DNA gyrase, and cell wall synthesis machinery [4]. Efflux pumps can also confer low-level resistance, which can facilitate the survival of bacteria in sub-lethal concentrations of antibiotics, thus providing an opportunity for the acquisition of additional resistance mechanisms [50]. Contributing to biofilm formation, efflux pumps also enhances resistance by creating a physical barrier that limits antibiotic penetration and protects the bacterial community [51].

We also discussed about other efflux pumps in page 7 line 153-162, as follows:

On the other hand, the MFS pumps are characterised by the transportation of a more limited range of substrates, including tetracyclines and chloramphenicol [52]. The SMR family pumps, which are smaller in size, facilitate the expulsion of smaller cationic compounds, such as quaternary ammonium compounds and dyes [53]. The ABC family, though primarily involved in nutrient transport, can also participate in antibiotic resistance [54]. Finally, the MATE family pumps, such as PmpM, use proton or sodium ion gradients to extrude substrates like fluoroquinolones and other cationic compounds [55]. The synergistic activity of these efflux pump families enables P. aeruginosa to achieve a high level of MDR, making it a challenging pathogen to treat in clinical settings.

How does horizontal gene transfer facilitate the acquisition of antibiotic resistance genes in P. aeruginosa, and what are the main genetic elements involved? (Lines 100–105)

Response:

Thank you for your great suggestions. We have added this part in Page 8, Line 187-193, as follows: Plasmids are key vectors for the spread of β-lactams, aminoglycosides, and fluoroquinolones resistance via conjugation [56]. Next, transposons, such as Tn21, facilitate the mobilisation of resistance genes by integrating into plasmids or the chromosome, while integrons capture and express resistance gene cassettes such as aadA and dfrA [57]. Lastly, genomic islands, such as PAGI-1 and PAPI-1, can also harbour anibiotic resistance and virulence genes, where they integrate into the chromosome and spread via conjugation or transduction [57].

What are the key mutations that contribute to adaptive resistance in P. aeruginosa, particularly in response to colistin and carbapenems? (Lines 107–120)

Response:

Thank you for your great suggestions. We have added this part in Page 9, Line 199, as follows:

Chromosomal β-Lactamase

ampC

overexpression of ampC, a chromosomal β-lactamase

hydrolyzes carbapenems, particularly when combined with oprD mutations.

[8]

pmrAB (PhoPQ) Two-Component System

pmrABphoPQ

lead to LPS modifications

decreasing colistin binding

[9]

Lipid A Biosynthesis Genes

lpxA, lpxC, and lpxD

lead to the complete loss of LPS production

resulting in colistin resistance through the absence of the drug's primary target

[10]

How does the quorum sensing (QS) system regulate biofilm formation, and what are the potential therapeutic targets within QS pathways? (Lines 147–152)

Response:

Thank you for your suggestion. We have added this part in Page 13-14, Line 257-272 as follows:

Firstly, four QS systems have been identified: las, rhl, pqs and iqs [58].The function of QS is to regulate population dynamics through the secretion of autoinducers (AIs). In the las system, LasI and LasR are mainly responsible for the synthesis and detection of N-(3-oxododecanoyl)-L-homoserine lactone (3-oxo-C12-HSL) respectively [59]. This autoinducer activates genes involved in biofilm formation, including those encoding EPS components [59]. The rhl system involves the transcription factor RhlR, which, upon reaching a threshold concentration, facilitates the expression of the lasI gene, in turn promoting the production of virulence factors, including LasA protease, LasB elastase, Apr alkaline protease, and exotoxin A. RhlI and RhlR are primarily responsible for the synthesis and detection of N-butanoyl-L-homoserine lactone (C4-HSL), which also promotes the transcription of RhlI, which in turn promotes the expression of virulence factors [59]. Meanwhile, Rhl system regulates the production of rhamnolipids, which are essential for biofilm structuring and dispersal [60]. The pqs system generates 2-heptyl-hydroxy-4-quinolone, which has the capacity to interact with the acylhomoserine lactone (AHL) system in a complex manner, thereby functioning as an intermediate hub between the las and rhl systems [61].

What advantages does phage therapy offer over conventional antibiotics in treating antibiotic-resistant P. aeruginosa, and what are its primary limitations? (Lines 211–261)

Response:

Thank you for your great suggestion. We have added the advantages in Page 19, Line 383-397, as follows:

The rationale behind utilising phage therapy is multi-faceted. Phages exhibit a high degree of penetration and specificity in comparison with conventional antibiotic therapies, which is useful in personalized treatment [62]. Furthermore, phages do not produce harm to human cells and resident microbes, thus achieving the effect of killing the target bacteria without interfering with the normal microbiota in the human body [63]. Next, phages are effective against MDR and XDR strains, providing a viable alternative where antibiotics fail. A further advantage of phage therapy is that some phages can penetrate and disrupt biofilms, via the production of EPS-degrading depolymerases [64]. Additionally, phages are able to self-replicate at the site of infection, ensuring elevated local concentrations, and are self-limiting when they are cleared once the bacterial host is eliminated [63].

For limitations, we discussed in Page 20 line 420-424, as follows:

As a new therapy under development, the primary limitations identified for this system include a restricted host range, necessitating precise strain matching; inadequate standardized protocols in managing pharmacodynamics (PD) and pharmacokinetics (PK) [65], and emergence of phage resistance.

How do antimicrobial peptides (AMPs) target P. aeruginosa, and what challenges exist in their clinical application? (Lines 305–334)

Response:

Thank you for your suggestions. We have added the new part in Page 25, Line 519-537, as follows:  

Although AMP holds great promise for the treatment of bacterial infections , the clinical use of AMPs is encumbered by several major challenges, the most significant of which pertain to their susceptibility to host and bacterial proteolytic enzymes, thereby affecting their stability and diminishing their therapeutic half-life [66]. In addition, despite their selectivity for bacterial membranes, certain AMPs exhibit haemolytic activity, which raises safety concerns. Thirdly, while the probability of bacterial resistance to antimicrobial peptides is comparatively lower than that of conventional antibiotics, P. aeruginosa may elicit resistance through mechanisms including membrane modification, efflux pumps and protease production [67],[68].

A further challenge is posed by the high production cost of synthetic AMPs, especially those with complex structures or those which contain unnatural amino acids [69]. Expression of AMP genes in microbes by genetic engineering may lead to self-killing of the host microbe [70]. Furthermore, AMPs frequently demonstrate suboptimal pharmacokinetic properties, such as rapid systemic clearance and limited bioavailability, necessitating the development of advanced delivery systems to enhance their efficacy [71]. Additionally, the host immune system may recognise AMPs as foreign substances, which could result in immune clearance or inflammatory responses [72].

What are the mechanisms by which quorum sensing inhibitors (QSIs) disrupt bacterial communication, and how can P. aeruginosa develop resistance to QSIs? (Lines 337–370)

Response:

Thank you for your suggestions. We have added the part on mechanisms in Page 26, Line 541-542, as follows:

QSIs disrupt bacterial communication by targeting key components of the quorum sensing (QS) system, which bacteria use to coordinate group behaviors [73]. The inhibition of QS can be achieved through the various approaches: (i) inhibition of the synthesis of QS signaling molecule, (ii) degradation of the QS signaling molecule, (iii) competition of QSI with the signaling molecule over the receptor binding site, (iv) inhibition of QS genes by preventing the signaling molecule from binding to their promoters, and (v) scavenging of QS signaling molecule via macromolecules such as antibodies and cyclodextrins [74, 75]. In terms of their chemical nature, QSIs can be classified as natural products, synthetic molecules and antibodies (Table 3). Since QSIs do not kill the target bacteria, QSIs are typically used in combination with antibiotics to improve antibiotic efficacy [76].

We also discussed the resistance to QSIs in Page 33 line 586-593:

The pathogen has the capacity to activate alternative signalling systems, such as the pqs QS, in order to compensate for inhibited QS circuits. Furthermore, P. aeruginosa may produce enzymes (e.g. oxidoreductases) that degrade or modify QSIs, rendering them ineffective [77]. Phenotypic heterogeneity within bacterial populations has also been demonstrated as a factor that can contribute to resistance, where QS-deficient "cheater" cells exploit QS-dependent public goods, such as virulence factors, without contributing to signal production, thereby reducing the selective pressure imposed by QSIs [77].

How can CRISPR-Cas systems be leveraged to combat antibiotic resistance in P. aeruginosa, and what challenges exist in their clinical application? (Lines 371–421)

Response:

Thank you for your great suggestions. We have actually included this part in Page 35-36, Line 638-656, as follows:

3.4.2 Progress and challenges in clinical application.

The CRISPR-Cas system has demonstrated considerable potential in addressing drug-resistant bacteria. However, the use of a solitary non-essential target for remediation proves inadequate when confronted with the complexity of MDR plasmids. The employment of CRISPR arrays or the utilisation of multiple sgRNAs, in conjunction with the design of CRISPR-Cas systems targeting essential genes on drug-resistant plasmids , can enhance the efficiency of CRISPR-Cas-based cleanup processes [78].

Next, the efficiency of delivery of the large 160kDa protein-RNA complex to the site of the pathogen and the subsequent exertion of antimicrobial effects remain a significant challenge. Existing literature proposes a range of methods for delivery, including the use of plasmid vectors [79], mild phages [80], nanoparticles [81], and electroporation [82]. However, these methods face challenges such as low stability, delivery efficiency, and potential immunological risks [81]. Moreover, CRISPR-Cas therapies are confronted with other technical challenges, including off-target effects, PAM sequence requirements [83], complex diversity of microbial communities; and even evolved resistance to the CRISPR Cas system [78]. Addressing these challenges is imperative for the widespread adoption and successful implementation of this technology.

What potential do nanoparticles have in overcoming drug resistance in P. aeruginosa, and how do they enhance drug delivery and efficacy? (Lines 422–452)

Response:

Thank you for your great suggestions. We have added the current first-line antibiotics used to treat multidrug-resistant Pseudomonas infections in Page 37, Line 693-704, as follows:

The application of nanoparticles has been shown to enhance the delivery and efficacy of drugs through multiple mechanisms. Nanoparticles can improve the solubility and stability of drugs that are poorly water-soluble, thereby increasing their bioavailability and therapeutic potential [84]. Through the modification of their surface with polymers such as polyethylene glycol (PEG), nanoparticles can reduce opsonization and extend their circulation time in the bloodstream, resulting in greater accumulation at infection sites [85]. The functionalization of nanoparticles with ligands, such as antibodies or peptides, enables targeted delivery of nanoparticles to P. aeruginosa with minimized off-target effects [86]. Furthermore, nanoparticles can overcome physiological barriers, such as the blood-brain barrier or epithelial layers, to deliver drugs to otherwise inaccessible infection sites [87].

References:

  1. Reynolds, D. and M. Kollef, The Epidemiology and Pathogenesis and Treatment of Pseudomonas aeruginosa Infections: An Update. Drugs, 2021. 81(18): p. 2117-2131.
  2. Micek, S.T., et al., An international multicenter retrospective study of Pseudomonas aeruginosa nosocomial pneumonia: impact of multidrug resistance. Critical Care, 2015. 19: p. 1-8.
  3. Theuretzbacher, U., Accelerating resistance, inadequate antibacterial drug pipelines and international responses. International journal of antimicrobial agents, 2012. 39(4): p. 295-299.
  4. Du, D., et al., Multidrug efflux pumps: structure, function and regulation. Nature Reviews Microbiology, 2018. 16(9): p. 523-539.
  5. Black, P.A., et al., Energy metabolism and drug efflux in Mycobacterium tuberculosis. Antimicrobial agents and chemotherapy, 2014. 58(5): p. 2491-2503.
  6. Di Pilato, V., et al., Identification of a Novel Plasmid Lineage Associated With the Dissemination of Metallo-β-Lactamase Genes Among Pseudomonads. Frontiers in Microbiology, 2019. 10.
  7. Yoon, E.-J. and S.H. Jeong, Mobile Carbapenemase Genes in Pseudomonas aeruginosa. Frontiers in Microbiology, 2021. 12.
  8. Jacoby George, A., AmpC β-Lactamases. Clinical Microbiology Reviews, 2009. 22(1): p. 161-182.
  9. Huang, J., et al., Regulating polymyxin resistance in Gram-negative bacteria: roles of two-component systems PhoPQ and PmrAB. Future Microbiol, 2020. 15(6): p. 445-459.
  10. Raetz, C.R. and C. Whitfield, Lipopolysaccharide endotoxins. Annu Rev Biochem, 2002. 71: p. 635-700.
  11. Varadarajan, A.R., et al., An integrated model system to gain mechanistic insights into biofilm-associated antimicrobial resistance in Pseudomonas aeruginosa MPAO1. npj Biofilms and Microbiomes, 2020. 6(1): p. 46.
  12. Wang, C., J. Wang, and Z. Mi, Pseudomonas aeruginosa producing VIM-2 metallo-β-lactamases and carrying two aminoglycoside-modifying enzymes in China. Journal of Hospital Infection, 2006. 62(4): p. 522-524.
  13. Cayci, Y.T., A. Coban, and M. Gunaydin, Investigation of plasmid-mediated quinolone resistance in Pseudomonas aeruginosa clinical isolates. Indian journal of medical microbiology, 2014. 32(3): p. 285-289.
  14. Cazares, A., et al., A megaplasmid family driving dissemination of multidrug resistance in Pseudomonas. Nature Communications, 2020. 11(1): p. 1370.
  15. Hernández-Ramírez, K.C., et al., A plasmid-encoded mobile genetic element from Pseudomonas aeruginosa that confers heavy metal resistance and virulence. Plasmid, 2018. 98: p. 15-21.
  16. Van Melderen, L. and M. Saavedra De Bast, Bacterial toxin–antitoxin systems: more than selfish entities? PLoS genetics, 2009. 5(3): p. e1000437.
  17. Catalano, A., et al., Multidrug resistance (MDR): A widespread phenomenon in pharmacological therapies. Molecules, 2022. 27(3): p. 616.
  18. Paul, V.D., et al., Lysis-deficient phages as novel therapeutic agents for controlling bacterial infection. BMC microbiology, 2011. 11: p. 1-9.
  19. Oechslin, F., et al., Synergistic interaction between phage therapy and antibiotics clears Pseudomonas aeruginosa infection in endocarditis and reduces virulence. The Journal of infectious diseases, 2017. 215(5): p. 703-712.
  20. Pletzer, D. and R.E. Hancock, Antibiofilm peptides: potential as broad-spectrum agents. Journal of bacteriology, 2016. 198(19): p. 2572-2578.
  21. Memariani, M. and H. Memariani, Anti-biofilm effects of melittin: lessons learned and the path ahead. International Journal of Peptide Research and Therapeutics, 2024. 30(3): p. 29.
  22. Gomes, D., et al., Pexiganan in combination with nisin to control polymicrobial diabetic foot infections. Antibiotics, 2020. 9(3): p. 128.
  23. Yin, C., et al., Advances in development of novel therapeutic strategies against multi-drug resistant Pseudomonas aeruginosa. Antibiotics, 2024. 13(2): p. 119.
  24. Ramamourthy, G. and H.J. Vogel, Antibiofilm activity of lactoferrin-derived synthetic peptides against Pseudomonas aeruginosa PAO1. Biochemistry and Cell Biology, 2021. 99(1): p. 138-148.
  25. Sahoo, A., et al., Antimicrobial peptides derived from insects offer a novel therapeutic option to combat biofilm: A review. Frontiers in microbiology, 2021. 12: p. 661195.
  26. Ali, W., et al., Host defence peptides: A potent alternative to combat antimicrobial resistance in the era of the covid-19 pandemic. Antibiotics, 2022. 11(4): p. 475.
  27. Liu, Q., et al., Thanatin: A Promising Antimicrobial Peptide Targeting the Achilles’ Heel of Multidrug-Resistant Bacteria. International Journal of Molecular Sciences, 2024. 25(17): p. 9496.
  28. Ruffin, M. and E. Brochiero, Repair process impairment by Pseudomonas aeruginosa in epithelial tissues: major features and potential therapeutic avenues. Frontiers in cellular and infection microbiology, 2019. 9: p. 182.
  29. Greco, I., et al., Characterization, mechanism of action and optimization of activity of a novel peptide-peptoid hybrid against bacterial pathogens involved in canine skin infections. Scientific Reports, 2019. 9(1): p. 3679.
  30. Rodrigues, M., et al., Conjugative delivery of CRISPR-Cas9 for the selective depletion of antibiotic-resistant enterococci. Antimicrobial Agents and Chemotherapy, 2019. 63(11): p. 10.1128/aac. 01454-19.
  31. El Husseini, N., J.A. Carter, and V.T. Lee, Urinary tract infections and catheter-associated urinary tract infections caused by Pseudomonas aeruginosa. Microbiology and Molecular Biology Reviews, 2024. 88(4): p. e00066-22.
  32. Chi, Y., et al., The efficacy and safety of Ceftolozane-Tazobactam in the treatment of GNB infections: a systematic review and meta-analysis of clinical studies. Expert Review of Anti-infective Therapy, 2023. 21(2): p. 189-201.
  33. Guzek, A., Z. Rybicki, and D. Tomaszewski, An Analysis of the Type and Antimicrobial Resistance of Carbapenemase-Producing Enterobacteriaceae Isolated at the Military Institute of Medicine in Warsaw, Poland, 2009-2016. Jundishapur Journal of Microbiology, 2019. 12(1): p. 1-6.
  34. Qin, S., et al., Pseudomonas aeruginosa: pathogenesis, virulence factors, antibiotic resistance, interaction with host, technology advances and emerging therapeutics. Signal transduction and targeted therapy, 2022. 7(1): p. 199.
  35. van Duin, D. and R.A. Bonomo, Ceftazidime/Avibactam and Ceftolozane/Tazobactam: Second-generation β-Lactam/β-Lactamase Inhibitor Combinations. Clin Infect Dis, 2016. 63(2): p. 234-41.
  36. Ito, A., et al., In Vitro Antibacterial Properties of Cefiderocol, a Novel Siderophore Cephalosporin, against Gram-Negative Bacteria. Antimicrob Agents Chemother, 2018. 62(1).
  37. Mikhail, S., et al., Evaluation of the Synergy of Ceftazidime-Avibactam in Combination with Meropenem, Amikacin, Aztreonam, Colistin, or Fosfomycin against Well-Characterized Multidrug-Resistant Klebsiella pneumoniae and Pseudomonas aeruginosa. Antimicrob Agents Chemother, 2019. 63(8).
  38. Torres, D.A., et al., Colistin resistance in Gram-negative bacteria analysed by five phenotypic assays and inference of the underlying genomic mechanisms. BMC Microbiology, 2021. 21(1): p. 321.
  39. Montini, G., et al., Short Oral Antibiotic Therapy for Pediatric Febrile Urinary Tract Infections: A Randomized Trial. Pediatrics, 2024. 153(1).
  40. Noronha, A.A., et al., Short- versus standard-course antibiotic therapy for urinary tract infection in children: a systematic review and meta-analysis. Pediatr Nephrol, 2024.
  41. Lashkar, M.O. and M.C. Nahata, Antimicrobial Pharmacotherapy Management of Urinary Tract Infections in Pediatric Patients. J Pharm Technol, 2018. 34(2): p. 62-81.
  42. Autore, G., et al., Management of Pediatric Urinary Tract Infections: A Delphi Study. Antibiotics (Basel), 2022. 11(8).
  43. Barken, K.B., et al., Roles of type IV pili, flagellum‐mediated motility and extracellular DNA in the formation of mature multicellular structures in Pseudomonas aeruginosa biofilms. Environmental microbiology, 2008. 10(9): p. 2331-2343.
  44. Mugunthan, S., et al., RNA is a key component of extracellular DNA networks in Pseudomonas aeruginosa biofilms. Nature Communications, 2023. 14(1): p. 7772.
  45. Costerton, J.W., et al., Microbial biofilms. 1995.
  46. Anderl, J.N., M.J. Franklin, and P.S. Stewart, Role of Antibiotic Penetration Limitation in <i>Klebsiella pneumoniae</i> Biofilm Resistance to Ampicillin and Ciprofloxacin. Antimicrobial Agents and Chemotherapy, 2000. 44(7): p. 1818-1824.
  47. Li, S., et al., Biofilm matrix cloaks bacterial quorum sensing chemoattractants from predator detection. The ISME Journal, 2022. 16(5): p. 1388-1396.
  48. Chan, S.Y., et al., Biofilm matrix disrupts nematode motility and predatory behavior. The ISME Journal, 2021. 15(1): p. 260-269.
  49. Liu, Y.S., et al., Dual-species proteomics and targeted intervention of animal-pathogen interactions. Journal of Advanced Research, 2024.
  50. Yamasaki, S., et al., Drug resistance and physiological roles of RND multidrug efflux pumps in Salmonella enterica, Escherichia coli and Pseudomonas aeruginosa. Microbiology (Reading), 2023. 169(6).
  51. Kumawat, M., et al., Role of bacterial efflux pump proteins in antibiotic resistance across microbial species. Microb Pathog, 2023. 181: p. 106182.
  52. Pasqua, M., et al., Host - Bacterial Pathogen Communication: The Wily Role of the Multidrug Efflux Pumps of the MFS Family. Front Mol Biosci, 2021. 8: p. 723274.
  53. Chung, Y.J. and M.H. Saier, Jr., SMR-type multidrug resistance pumps. Curr Opin Drug Discov Devel, 2001. 4(2): p. 237-45.
  54. Greene, N.P., et al., Antibiotic Resistance Mediated by the MacB ABC Transporter Family: A Structural and Functional Perspective. Front Microbiol, 2018. 9: p. 950.
  55. He, G.X., et al., EmmdR, a new member of the MATE family of multidrug transporters, extrudes quinolones from Enterobacter cloacae. Arch Microbiol, 2011. 193(10): p. 759-65.
  56. Subedi, D., A.K. Vijay, and M. Willcox, Overview of mechanisms of antibiotic resistance in Pseudomonas aeruginosa: an ocular perspective. Clin Exp Optom, 2018. 101(2): p. 162-171.
  57. Morales-Espinosa, R., et al., Genetic and Phenotypic Characterization of a Pseudomonas aeruginosa Population with High Frequency of Genomic Islands. PLOS ONE, 2012. 7(5): p. e37459.
  58. Rasamiravaka, T. and M. El Jaziri, Quorum-Sensing Mechanisms and Bacterial Response to Antibiotics in P. aeruginosa. Current Microbiology, 2016. 73(5): p. 747-753.
  59. Bjarnsholt, T. and M. Givskov, The role of quorum sensing in the pathogenicity of the cunning aggressor Pseudomonas aeruginosa. Analytical and Bioanalytical Chemistry, 2007. 387(2): p. 409-414.
  60. Rasamiravaka, T., et al., Pseudomonas aeruginosa Biofilm Formation and Persistence, along with the Production of Quorum Sensing-Dependent Virulence Factors, Are Disrupted by a Triterpenoid Coumarate Ester Isolated from Dalbergia trichocarpa, a Tropical Legume. PLoS One, 2015. 10(7): p. e0132791.
  61. Wilder, C.N., S.P. Diggle, and M. Schuster, Cooperation and cheating in Pseudomonas aeruginosa: the roles of the las, rhl and pqs quorum-sensing systems. The ISME Journal, 2011. 5(8): p. 1332-1343.
  62. Leitner, L., et al., Intravesical bacteriophages for treating urinary tract infections in patients undergoing transurethral resection of the prostate: a randomised, placebo-controlled, double-blind clinical trial. The Lancet Infectious Diseases, 2021. 21(3): p. 427-436.
  63. Vaitekenas, A., et al., Pseudomonas aeruginosa Resistance to Bacteriophages and Its Prevention by Strategic Therapeutic Cocktail Formulation. Antibiotics, 2021. 10(2): p. 145.
  64. Kushwaha, S.O., et al., Bacteriophages as a potential substitute for antibiotics: A comprehensive review. Cell Biochem Funct, 2024. 42(3): p. e4022.
  65. Nang, S.C., et al., Pharmacokinetics/pharmacodynamics of phage therapy: a major hurdle to clinical translation. Clin Microbiol Infect, 2023. 29(6): p. 702-709.
  66. Aoki, W. and M. Ueda, Characterization of Antimicrobial Peptides toward the Development of Novel Antibiotics. Pharmaceuticals (Basel), 2013. 6(8): p. 1055-81.
  67. Costa, F., et al., Clinical Application of AMPs, in Antimicrobial Peptides: Basics for Clinical Application, K. Matsuzaki, Editor. 2019, Springer Singapore: Singapore. p. 281-298.
  68. Luo, Y. and Y. Song, Mechanism of Antimicrobial Peptides: Antimicrobial, Anti-Inflammatory and Antibiofilm Activities. International Journal of Molecular Sciences, 2021. 22(21): p. 11401.
  69. Lei, J., et al., The antimicrobial peptides and their potential clinical applications. Am J Transl Res, 2019. 11(7): p. 3919-3931.
  70. Song, Y.-Q., et al., Effects of synthetic peptide RP557 and its origin, LL-37, on carbapenem-resistant Pseudomonas aeruginosa. Microbiology Spectrum, 2023. 11(5): p. e00430-23.
  71. Mwangi, J., et al., Design methods for antimicrobial peptides with improved performance. Zool Res, 2023. 44(6): p. 1095-1114.
  72. Huan, Y., et al., Antimicrobial Peptides: Classification, Design, Application and Research Progress in Multiple Fields. Front Microbiol, 2020. 11: p. 582779.
  73. O'Loughlin, C.T., et al., A quorum-sensing inhibitor blocks Pseudomonas aeruginosa virulence and biofilm formation. Proc Natl Acad Sci U S A, 2013. 110(44): p. 17981-6.
  74. Castang, S., et al., N-Sulfonyl homoserine lactones as antagonists of bacterial quorum sensing. Bioorganic & Medicinal Chemistry Letters, 2004. 14(20): p. 5145-5149.
  75. Kalia, V.C., Quorum sensing inhibitors: An overview. Biotechnology Advances, 2013. 31(2): p. 224-245.
  76. Tan, S.Y.-Y., et al., Identification of Five Structurally Unrelated Quorum-Sensing Inhibitors of Pseudomonas aeruginosa from a Natural-Derivative Database. Antimicrobial Agents and Chemotherapy, 2013. 57(11): p. 5629-5641.
  77. Kalia, V.C., T.K. Wood, and P. Kumar, Evolution of resistance to quorum-sensing inhibitors. Microb Ecol, 2014. 68(1): p. 13-23.
  78. Gholizadeh, P., et al., Role of CRISPR-Cas system on antibiotic resistance patterns of Enterococcus faecalis. Annals of clinical microbiology and antimicrobials, 2021. 20: p. 1-12.
  79. García-Gutiérrez, E., et al., CRISPR content correlates with the pathogenic potential of Escherichia coli. PloS one, 2015. 10(7): p. e0131935.
  80. Tao, S., et al., The application of the CRISPR-Cas system in antibiotic resistance. Infection and drug resistance, 2022: p. 4155-4168.
  81. Wan, F., et al., Novel strategy to combat antibiotic resistance: a sight into the combination of CRISPR/Cas9 and nanoparticles. Pharmaceutics, 2021. 13(3): p. 352.
  82. Chen, Z., et al., CRISPR/Cas12a and immuno-RCA based electrochemical biosensor for detecting pathogenic bacteria. Journal of Electroanalytical Chemistry, 2021. 901: p. 115755.
  83. Wang, J., et al., Engineering a PAM-flexible SpdCas9 variant as a universal gene repressor. Nature communications, 2021. 12(1): p. 6916.
  84. Sguizzato, M., et al., Bilosomes and Biloparticles for the Delivery of Lipophilic Drugs: A Preliminary Study. Antioxidants (Basel), 2023. 12(12).
  85. Suk, J.S., et al., PEGylation as a strategy for improving nanoparticle-based drug and gene delivery. Adv Drug Deliv Rev, 2016. 99(Pt A): p. 28-51.
  86. Hemeg, H.A., Nanomaterials for alternative antibacterial therapy. Int J Nanomedicine, 2017. 12: p. 8211-8225.
  87. Zhu, Q., et al., Oral delivery of proteins and peptides: Challenges, status quo and future perspectives. Acta Pharm Sin B, 2021. 11(8): p. 2416-2448.

Round 2

Reviewer 2 Report

Comments and Suggestions for Authors

I have no further comments. It would be helpful to include a version with the changes highlighted to facilitate the review process.

Author Response

Comment 1: I have no further comments. It would be helpful to include a version with the changes highlighted to facilitate the review process.

Response 1: Thank you for your suggestion. We had included a version with changes highlighted in red, so please see the attachment.
